# Phage resistance profiling identifies new genes required for biogenesis and modification of the corynebacterial cell envelope

Amelia C McKitterick[1,2], Thomas G Bernhardt[1,2]*

[1]Department of Microbiology, Harvard Medical School, Boston, United States; [2]Howard Hughes Medical Institute, Chevy Chase, United States

**Abstract** Bacteria of the order Corynebacteriales including pathogens such as *Mycobacterium tuberculosis* and *Corynebacterium diphtheriae* are characterized by their complex, multi-layered envelope. In addition to a peptidoglycan layer, these organisms possess an additional polysaccharide layer made of arabinogalactan and an outer membrane layer composed predominantly of long-chain fatty acids called mycolic acids. This so-called mycolata envelope structure is both a potent barrier against antibiotic entry into cells and a target of several antibacterial therapeutics. A better understanding of the mechanisms underlying mycolata envelope assembly therefore promises to reveal new ways of disrupting this unique structure for the development of antibiotics and antibiotic potentiators. Because they engage with receptors on the cell surface during infection, bacteriophages have long been used as tools to uncover important aspects of host envelope assembly. However, surprisingly little is known about the interactions between Corynebacteriales phages and their hosts. We therefore made use of the phages Cog and CL31 that infect *Corynebacterium glutamicum* (*Cglu*), a model member of the Corynebacteriales, to discover host factors important for phage infection. A high-density transposon library of *Cglu* was challenged with these phages followed by transposon sequencing to identify resistance loci. The analysis identified an important role for mycomembrane proteins in phage infection as well as components of the arabinogalactan and mycolic acid synthesis pathways. Importantly, the approach also implicated a new gene (*cgp_0396*) in the process of arabinogalactan modification and identified a conserved new factor (AhfA, Cpg_0475) required for mycolic acid synthesis in *Cglu*.

*For correspondence:
thomas_bernhardt@hms.harvard.edu

Competing interest: The authors declare that no competing interests exist.

## Editor's evaluation

The authors perform a Transposon-Sequencing screen to determine bacterial factors (including receptors) important for infection by two phages in the model bacterium Corynebacterium glutamicum. Using their established high-density transposon library, they identify genes required for infection with the phages Cog and CL31. They also identified a spontaneous phage-resistant mutant that led to the discovery of a gene involved in mycolic acid synthesis. Overall, the work is of broad interest to scientists in the field of cell wall biogenesis, phage infection, and bacterial cell biology.

## Introduction

The envelope surrounding bacteria is essential for the structural integrity of the cell in addition to serving as the interface with the environment. Members of the Corynebacteriales order, including pathogens such as *Mycobacterium tuberculosis* (*Mtb*) and *Corynebacterium diphtheriae*, have a

particularly complex surface architecture referred to as the mycolata envelope (*Figure 1A*; *Jankute et al., 2015*; *Daffé and Marrakchi, 2019*). Like most bacteria, the cytoplasmic membrane is fortified with a cell wall made of the crosslinked heteropolymer peptidoglycan (PG). However, unique to these organisms are the additional layers attached to the PG matrix. Namely, branched chains of arabinogalactan (AG) are covalently linked to the PG, and the AG layer is further modified through the esterification of long acyl chains of mycolic acids. Free mycolic acid glycolipids linked to trehalose in the form of trehalose monomycolate (TMM) and trehalose dimycolate (TDM) as well as other glycolipid species join with the covalently linked mycolic acids to form a mycomembrane outer layer that includes embedded proteins and is thought to be the functional equivalent of the outer membrane of Gram-negative bacteria.

Many of the drugs used in the treatment of *Mtb* infections target proteins involved in envelope biogenesis (*Abrahams and Besra, 2018*; *Dulberger et al., 2020*). The envelope also serves as a barrier to the entry of certain antibiotics. Therefore, in addition to providing fundamental insights into the growth and morphogenesis of the Corynebacteriales, studies of the mechanisms of envelope assembly in these organisms also promises to identify new vulnerabilities that will enable the development of novel therapies effective against mycobacterial pathogens. *Corynebacterium glutamicum* (*Cglu*) is an excellent model organism for studies of mycolate envelope biogenesis due to its rapid doubling time and relatively small genome. Additionally, unlike mycobacterial models, mutants lacking the mycomembrane or arabanan chains of the AG layer are viable (*Portevin et al., 2004*; *Alderwick et al., 2005*), simplifying the genetic analysis of envelope assembly.

Viruses that infect bacteria, called bacteriophages or simply phages, have long served as useful probes of envelope assembly in addition to being interesting biological entities in their own right. To initiate infection, phages require specific cell surface receptors, which can be proteins, glycolipids, or polysaccharides (*Bertozzi Silva et al., 2016*; *Dunne et al., 2018*). Phages engage with these receptors through both reversible and irreversible interactions and often require a combination of receptors to fully engage the cell surface and undergo the conformational changes necessary to penetrate the envelope. Host factors have additionally been implicated in facilitating the phage translocation

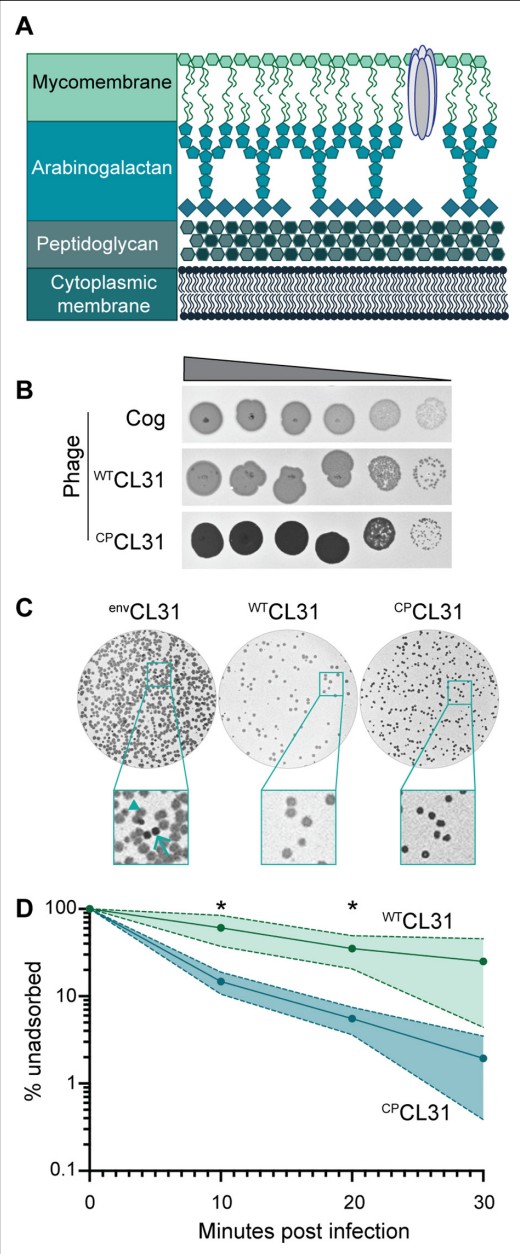

**Figure 1.** The *Corynebacterium glutamicum* (*Cglu*) envelope and adaptation of corynephages to the lab strain. (**A**) Cartoon of the multi-layered cell envelope of *Cglu* and other members of the Corynebacteriales. The mycomembrane is composed of mycolic acids covalently attached to the arabinogalactan layer or conjugated with trehalose. (**B**) Phages adapted to the MB001 lab strain of *Cglu*. Ten-fold serial dilutions of lysates for the indicated phages were prepared and 3 µL of each dilution spotted on a lawn of MB001. Large zones of growth inhibition are observed at high phage concentrations and individual plaques are observed at the lowest concentrations. (**C**) The CL31 phage forms plaques with different morphologies. (Left) Phage grown on the environmental host strain (envCL31) were plated on MB001 and formed both turbid

*Figure 1 continued*

(arrowhead) and clear (arrow) plaques. Both plaque types were purified and replated on MB001, retaining their turbid (WTCL31) or clear phenotype (CPCL31). The CPCL31 phage were found to have a mutation in a gene encoding a tail protein. (**D**) Phage adsorption assay. The fraction of unadsorbed WTCL31 and CPCL31 phages was determined at the indicated timepoint after mixing with MB001. Shaded regions indicate standard deviation and solid lines indicate the mean from three independent experiments. Significance was determined by two-tailed t test. *p<0.05.

The online version of this article includes the following source data and figure supplement(s) for figure 1:

**Source data 1.** Single nucleotide polymorphisms (SNPs) detected in phages adapted to MB001.

**Source data 2.** Adsorption source data.

**Figure supplement 1.** Adaptation of Cog phage to lab *Corynebacterium glutamicum* (*Cglu*) strain MB001 and genome sequencing.

through the layers of the cell envelope and into the cytoplasm where DNA is injected for replication (*Scandella and Arber, 1974*; *Cumby et al., 2015*; *Wenzel et al., 2020*), demonstrating that there are multiple steps involved in initiation of phage infection beyond engagement with the primary host receptor(s). Studies of phage-host interactions at the cell surface have been particularly helpful in uncovering mechanisms involved in envelope biogenesis. For example, the identification of the *Escherichia coli* porin LamB as the lambda phage receptor provided a powerful genetic system for investigating the process of protein transport from the cytoplasm to the outer membrane (*Randall-Hazelbauer and Schwartz, 1973*; *Szmelcman and Hofnung, 1975*; *Emr et al., 1978*). Similarly, the inner membrane mannose transporter subunits IIC$^{Man}$ and IID$^{Man}$ of *E. coli* were also demonstrated to be required for lambda phage DNA injection and have been used to investigate the phosphoenolpyruvate:-sugar phosphotransferase system (*Williams et al., 1986*; *Esquinas-Rychen and Erni, 2001*).

Surprisingly little is known about the receptor requirements for phages that infect members of the Corynebacteriales. The few studies that have been undertaken have hinted at roles of lipids in facilitating phage infection (*Furuchi and Tokunaga, 1972*; *Chen et al., 2009*), but the essential role of the mycomembrane in most Corynebacteriales has made further investigation challenging. We therefore used two phages, Cog and CL31, that infect *Cglu* to probe envelope assembly in the Corynebacteriales. A high-density *Cglu* transposon library was challenged with each phage to identify genes required for phage sensitivity. This analysis led to the discovery of new factors involved in mycolic acid synthesis and AG modification that are conserved in *Corynebacterial* pathogens, highlighting the utility of phage resistance profiling for enhancing our understanding of the mechanisms underlying mycolata envelope biogenesis.

## Results

### Adaptation of environmental corynephages to the laboratory *Cglu* strain MB001

The corynephages CL31 and Cog were obtained from the Félix d'Hérelle Reference Center for Bacterial Viruses along with a susceptible environmental corynebacterial host strain for each (ATCC 15990 and LP-6, respectively). The phages were then adapted to form plaques on the lab strain MB001, and the genomes of the adapted phages were sequenced (*Figure 1B*, *Figure 1—figure supplement 1A–B*). Notably, the lab-adapted variants of both Cog and CL31 had mutations in genes encoding putative tail proteins, suggesting that the surface of MB001 differs in some way from the environmental hosts necessitating a change in the phage particle for effective engagement (*Figure 1—source data 1*). In the case of Cog, the mutations were found in the initial lab-adapted isolate whereas for CL31, the mutations were associated with a change in plaque morphology. Lysates of CL31 grown on the environmental host (envCL31) formed plaques with two different morphologies on the lab host (*Figure 1C*). Most of the plaques were large and turbid, but a small percentage were small and clear. A similar heterogeneity in plaque morphology for lab-adapted CL31 was recently reported, but the cause of the phenomenon was not investigated (*Hünnefeld et al., 2021*). Phage purified from the clear plaques retained this phenotype and gave rise to only clear plaques on the MB001 host (*Figure 1B*).

Whole-genome sequencing of the clear plaque CL31 isolate (CPCL31) identified single nucleotide polymorphisms (SNPs) in the 3′ end of *clg55*, which encodes a predicted tail protein in the structural

protein locus of the genome (**Figure 1—source data 1**). Notably, previous work has identified a putative variable region within Clg55, independent of the region in which these mutations occur (**Hünnefeld et al., 2021**), suggesting that *clg55* is under selection, possibly due to its role in receptor recognition. Four additional ^CPCL31 variants were isolated, and Sanger sequencing revealed that all also had mutations in *clg55*, further implicating this structural gene in the altered plaque morphology phenotype (**Figure 1—source data 1**). To understand how changes in Clg55 influence plaque morphology, the ability of the different phages to adsorb to *Cglu* was assessed. Adsorption assays comparing ^WTCL31 with a ^CPCL31 variant indicated that the mutant forming clear plaques binds more efficiently to MB001, suggesting that the changes in Clg55 promote better engagement of the phage with a receptor or co-receptor on the surface of *Cglu* (**Figure 1D**), thereby leading to more efficient infection and the clear plaque phenotype. Structural predictions of the C-terminus of Clg55 detected remote homology with C1q-like domains involved in eukaryotic immune recognition by engaging with ligands such as lipopolysaccharide. Thus, Clg55 is a good candidate for the receptor binding protein of CL31.

## Transposon sequencing reveals host requirements for phage infection

With MB001-adapted corynephages in hand, we were positioned to use our recently constructed high-density transposon library of MB001 to perform a global genetic analysis for the identification of host factors required for phage infection (**Lim et al., 2019**). To this end, the library was challenged with a high multiplicity of infection (MOI = 5) of lab-adapted Cog, ^WTCL31, or ^CPCL31 in two independent replicates. After allowing time for phage adsorption, the challenged libraries, along with an untreated control sample, were plated on brain-heart infusion (BHI) agar. Colonies from each sample were then separately scraped and pooled, and genomic DNA (gDNA) was prepared from the resulting cell suspensions. Transposon sequencing (Tn-Seq) was then used to analyze the

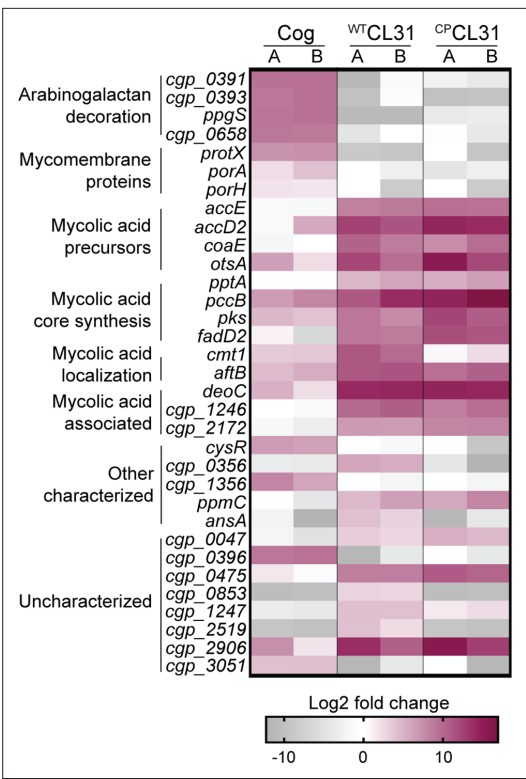

**Figure 2.** CL31 and Cog have distinct requirements for host envelope components. The heatmap shows genes that displayed a $\log_2$ fold change in transposon insertion reads of $\geq 5$ in the transposon sequencing (Tn-seq) analysis following challenge with the indicated phage. Each phage challenge was performed independently twice, and the $\log_2$ fold changes of each challenge are shown in the A and B columns, respectively. Genes were clustered based on predicted or published functions, which are indicated on the left. Depletion in transposon insertion reads is unlikely to be meaningful given that the experiment was a selection for phage resistance and that transposon mutants in genes that do not promote survival when inactivated are expected to be depleted by phage killing.

The online version of this article includes the following source data for figure 2:

**Source data 1.** Phage challenge transposon sequencing (Tn-Seq) metadata.

**Source data 2.** Transposon sequencing results.

**Source data 3.** Transposon enrichment source data.

profile of insertion mutants remaining in each of the libraries following phage challenge (**Figure 2— source data 1**). Mutations that block phage infection were expected to promote survival following phage treatment. Therefore, relative to the profile of the untreated control, we anticipated that the Tn-Seq profiles of phage-treated samples would be enriched for transposon insertions in genes encoding factors involved in the synthesis and/or localization of phage receptor(s) as well as other genes required for infection. Indeed, in the profiles of the phage challenged samples, transposon insertions were concentrated in just a small collection of genes (**Figure 2**), which were highly reproducible bewteen replicates (r=0.877). Notably, there was little overlap in the set of genes that were

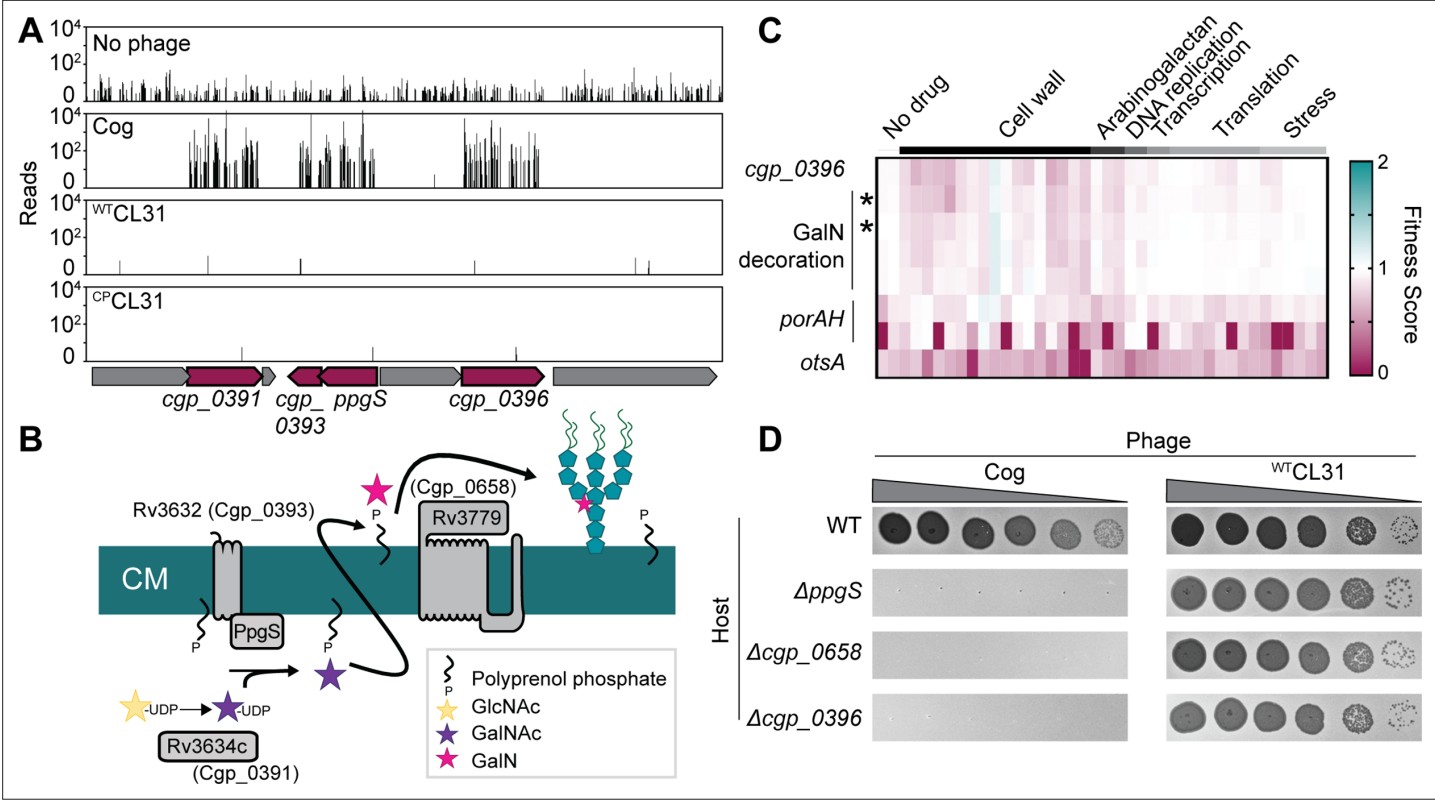

**Figure 3.** Cog requires genes implicated in arabinogalactan modification to infect *Corynebacterium glutamicum* (*Cglu*). (**A**) Transposon sequencing (Tn-Seq) insertion profiles for a locus encoding genes with potential functions in the modification of the *Cglu* arabinogalactan with galactosamine (GalN). Genes enriched for insertions following Cog infection are colored in burgundy. (**B**) Schematic showing the arabinogalactan modification pathway characterized in *Mycobacterium tuberculosis* (*Mtb*). The names of the corresponding *Cglu* homologs are given in parentheses. (**C**) Heatmap showing the previously determined phenotypic profiles of the indicated genes following challenge of the transposon library with a variety of antibiotics and other stressors. The target categories of antibiotics used for the challenge are indicated above the heatmap. Note the similarity of the profile for *cgp_0396* to that of the gene implicated in GalN modification of arabinogalactan, including the glycosyltransferases *ppgS* and *cgp_0658*, indicated with an asterisk. These genes are not clustered in the phenotypic profile near other genes like *porAH*, or *otsA*, which have different phenotypic fingerprints that are shown for comparison. Similarity determined by correlation, *$r$>0.9. Data from ***Sher et al., 2020***. (**D**) Validation of the Tn-Seq results for the predicted arabinogalactan modification genes. Ten-fold serial dilutions of lysates for the indicated phages were prepared and 3 µL of each dilution were spotted on a lawn of the indicated host strain.

The online version of this article includes the following source data and figure supplement(s) for figure 3:

**Source data 1.** Phenotypic profiling data.

**Figure supplement 1.** Transposon sequencing (Tn-Seq) insertion profiles for the *cgp_0658* locus with and without Cog challenge.

strongly enriched for insertions when comparing profiles from the Cog- and CL31-treated samples (***Figure 2***). Thus, the two phages appear to require a distinct set of factors for the efficient infection of *Cglu*. Conversely, there was more overlap between the two variants of CL31, with only a few genes having a differential enrichment profile. The results for each phage challenge experiment will be presented in turn below along with the new insights into cell envelope biogenesis gained from them.

## Cog phage requires AG modification and mycomembrane proteins for infection

Transposon insertions in genes implicated in the modification of the AG layer and genes encoding mycomembrane proteins were among the most highly enriched following challenge with Cog phage (***Figure 2***). Insertions in *ppgS*, *cgp_0658*, *cgp_0391*, *cgp_0393*, and *cgp_0396* were the most highly enriched in the Tn-Seq profile from Cog challenged cells (***Figures 2 and 3A***, ***Figure 3—figure supplement 1***). Several of these genes are related to those from *Mtb* that have been implicated in decorating the AG layer with a galactosamine (GalN) modification that has been implicated in *Mtb* pathogenesis

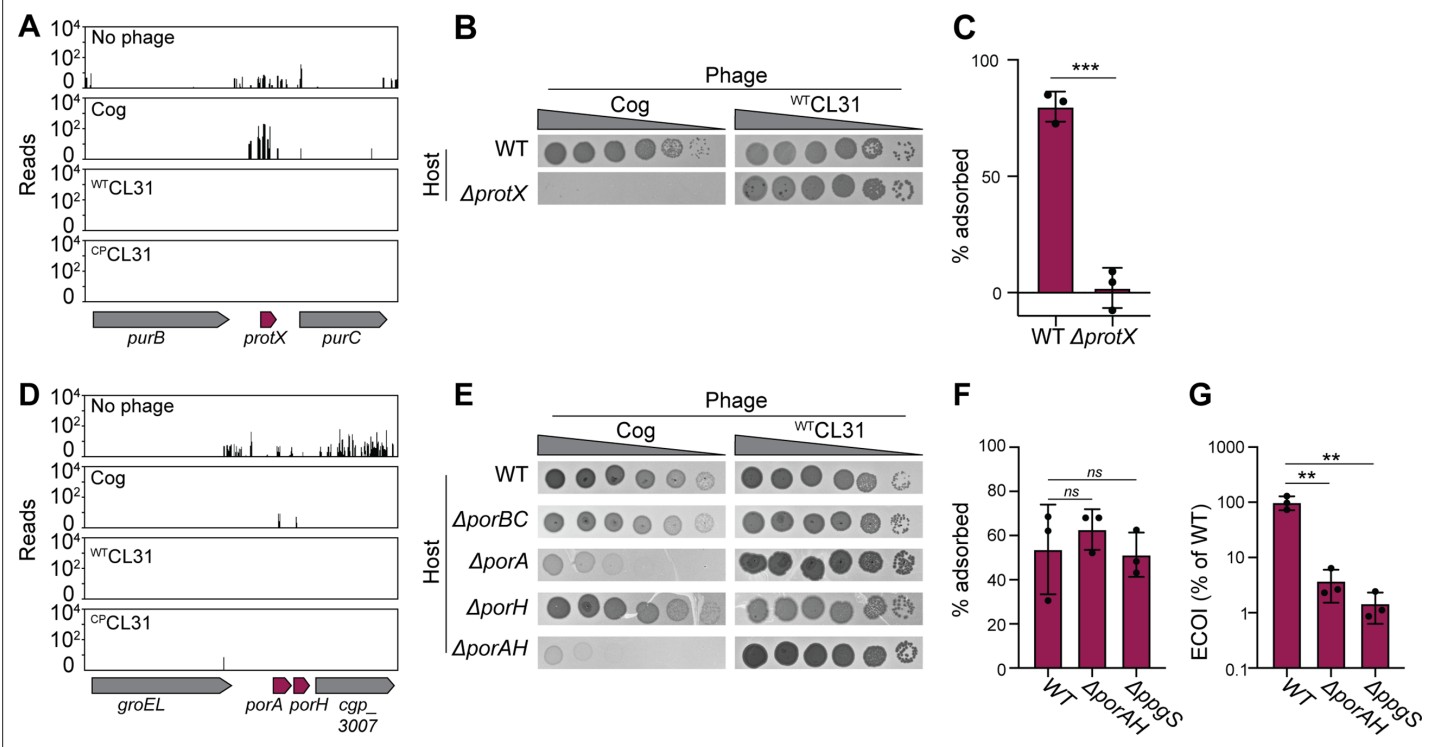

**Figure 4.** Mycomembrane proteins are required for Cog infection. (**A,D**) Transposon sequencing (Tn-Seq) insertion profile for the *protX* locus and the *porAH* locus, respectively. (**B,E**) Validation of the Tn-Seq results. Ten-fold serial dilutions of lysates for the indicated phages were prepared and 3 μL of each dilution were spotted on a lawn of the indicated host strain. (**C,F**) Percent of Cog adsorbed after 30 min of incubation with the indicated *Corynebacterium glutamicum* (*Cglu*) strain. (**G**). Efficiency by which Cog formed centers of infection (ECOI) on a lawn of wild-type cells following 15 min of adsorption to the indicated host cells. Circles represent individual data points, bar height represents mean, and error bars indicate standard deviation of three independent experiments. Significance was determined by two-tailed t tests. **p<0.005, ***p<0.001, *ns*, not significant.

The online version of this article includes the following source data for figure 4:

**Source data 1.** Adsorption and COI data.

(*Figure 3B*; *Draper et al., 1997*; *Skovierová et al., 2010*; *Wheat et al., 2015*). A recent analysis of the AG layer of *Cglu* found that it contains small amounts of GalN, suggesting that *Cglu* modifies its AG using a similar pathway to that found in *Mtb* (*Marchand et al., 2012*). Cgp_0391 is homologous to Rv3634c, which has been demonstrated in vitro to be a UDP-Gal/GalNAc-epimerase that catalyzes the interconversion of UDP-*N*-acetylglucosamine (GlcNAc) and UDP-*N*-acetylgalactosamine (GalNAc) (*Pardeshi et al., 2017*). The GalNAc sugar is transferred from UDP to the polyprenyl-phosphate (PolyP-P) lipid carrier by $^{Mtb}$PpgS with the assistance of Rv3632 (*Skovierová et al., 2010*), which is a homolog of Cgp_0393. The lipid linked sugar is then flipped and deacetylated by as yet unidentified factors to expose a GalN moiety to the extracellular surface of the membrane where it is then transferred to the AG polymer by Rv3779, which is an ortholog of Cgp_0658. The Cgp_0396 protein does not have a homolog in the *Mtb* AG modification system; however, it is a member of the GT2 family of glycosyltransferases, and mutants inactivating this protein have a phenotypic profile that clusters with the other *Cglu* factors predicted to function in AG modification (*Sher et al., 2020*; *Figure 3C*). Thus, Cgp_0396 is also likely to play a role in AG modification in *Cglu*. To test the role of the putative AG modification system in Cog infection, strains deleted for the glycosyltransferases *ppgS, cgp_658,* or *cgp_0396* were constructed. Cog was unable to form plaques on these strains even at very high phage concentrations, whereas the plaquing efficiency of the CL31 phage was unaffected by these mutations (*Figure 3D*). We therefore infer that AG modification is required for Cog infection and that Cgp_0396 is likely to be involved in the AG modification process.

In order to gain access to the AG layer, the Cog phage must first penetrate the mycomembrane surface layer. In addition to genes implicated in AG modification, transposon insertions in the poorly characterized gene *protX* and porin-encoding genes *porA* and *porH* were also highly enriched following

Cog challenge (*Figures 2 and 4*). Porins are proteins embedded in the outer cell membrane that allow nutrients and other molecules to cross this surface layer. In Gram-negative bacteria, beta-barrel porins are often used as receptors for phages (*Bertozzi Silva et al., 2016*). For example, the coliphages T4 and lambda use the porins OmpC and LamB, respectively, as receptors (*Randall-Hazelbauer and Schwartz, 1973*; *Yu and Mizushima, 1982*). Rather than adopting a beta-barrel structure, the characterized corynebacterial porins in the mycomembrane are small alpha-helical proteins (*Ziegler et al., 2008*). To date, four total porins have been identified, including PorA, PorB, PorC, and PorH (*Lichtinger et al., 2001*; *Costa-Riu et al., 2003*; *Hünten et al., 2005*), with PorA being thought of as the predominant protein in the mycomembrane (*Marchand et al., 2012*). Little is known about ProtX except that it is mycoloylated, a post-translational modification that is associated with proteins in the mycomembrane (*Huc et al., 2010*; *Kavunja et al., 2016*). Similar to the porins, ProtX is very small and predicted to be alpha-helical, suggesting it may also be a porin embedded in the mycomembrane.

We first analyzed the role of ProtX due to the high enrichment of transposon insertions in the gene encoding it following Cog challenge (*Figures 2 and 4A*). Loss of *protX* led to inhibition of Cog plaque formation (*Figure 4B*). Furthermore, the Cog phage was unable to adsorb to a *ΔprotX* host, suggesting that ProtX is a required receptor for Cog infection (*Figure 4C*). Notably, plaques were observed on the *protX* deletion strain at a low frequency (*Figure 4B*), suggesting that they were derived from phage mutants capable of using an additional or alternative receptor.

We next investigated the role of the *porA* and *porH* genes in Cog infection (*Figure 4D*). PorA and PorH were previously shown to form hetero-oligomeric (PorAH) pores in membranes in vitro with PorA alone also being capable of forming pores in the purified system (*Barth et al., 2010*). The genes encoding these porins are located adjacent to one another in the *Cglu* genome and form an apparent operon (*Figure 4D*). Deletion of both *porA* and *porH* as well as the deletion of *porA* alone was found to severely reduce the plating efficiency of Cog (*Figure 4E*), albeit to a lesser degree than *ΔprotX* (*Figure 4B*). Inactivation of *porH* alone did not affect plaque formation by Cog (*Figure 4E*), suggesting that this gene was a hit in the Tn-Seq analysis due to polar effects of *porH* insertions on *porA* expression. However, the deletion of *porAH* had a more severe effect on Cog plaque formation than the inactivation of *porA* alone (*Figure 4E*), suggesting some contribution of PorH to Cog infection, likely in the context of hetero-oligomers with PorA. Deletion of the *porB* and *porC* porin-encoding genes, which are also adjacent to one another on the chromosome, did not interfere with *Cglu* infection by Cog (*Figure 4E*). Thus, the phage appears to have a specific requirement for the PorA porin to promote the efficient infection of *Cglu*. Consistent with the Tn-Seq results, neither the *protX* deletion nor any of the porin gene deletions negatively affected the plaquing efficiency of the CL31 phage (*Figure 4B and E*). Therefore, unlike Cog, the CL31 phage does not require a specific porin to infect *Cglu*.

Surprisingly, unlike *ΔprotX* cells (*Figure 4C*), no significant defects in phage adsorption were observed for mutants lacking the PorA and PorH porins or the putative AG modification system (*Figure 4F*). However, analysis of the ability of the phages to successfully infect their hosts following adsorption by monitoring the ability to form a plaque (infectious center) on a lawn of a permissive wild-type host showed a significant drop in the number of infectious centers formed by Cog following adsorption to either of the mutant hosts compared to wild-type (*Figure 4G*). Despite the high enrichment of transposon insertions following phage challenge, the results indicate that neither PorAH nor decorations of the AG are essential for the phage to engage with the cell surface. Instead, these factors appear to be necessary for Cog infection post-attachment to ProtX, possibly by facilitating genome injection, similar to the inner membrane transport channels used by lambda and HK97 for DNA injection following attachment (*Scandella and Arber, 1974*; *Cumby et al., 2015*).

## Mycolic acids are required for CL31 infection

The collection of genes enriched in insertions following CL31 challenge differed significantly from those that protected against Cog infection (*Figure 2*). Rather than porin or AG modification, the inactivated genes that were enriched encode factors involved in mycolic acid synthesis or protein mycoloylation (*Figures 2 and 5A*). Mycolic acid synthesis begins in the cytoplasm where acyl chains contributed by FadD2 and the ACC complex are condensed by Pks to form an extended lipid moiety (*Portevin et al., 2004*; *Portevin et al., 2005*). This long-chain lipid is then attached to trehalose, which is primarily synthesized by the OtsAB pathway (*Murphy et al., 2005*). The resulting TMM precursor

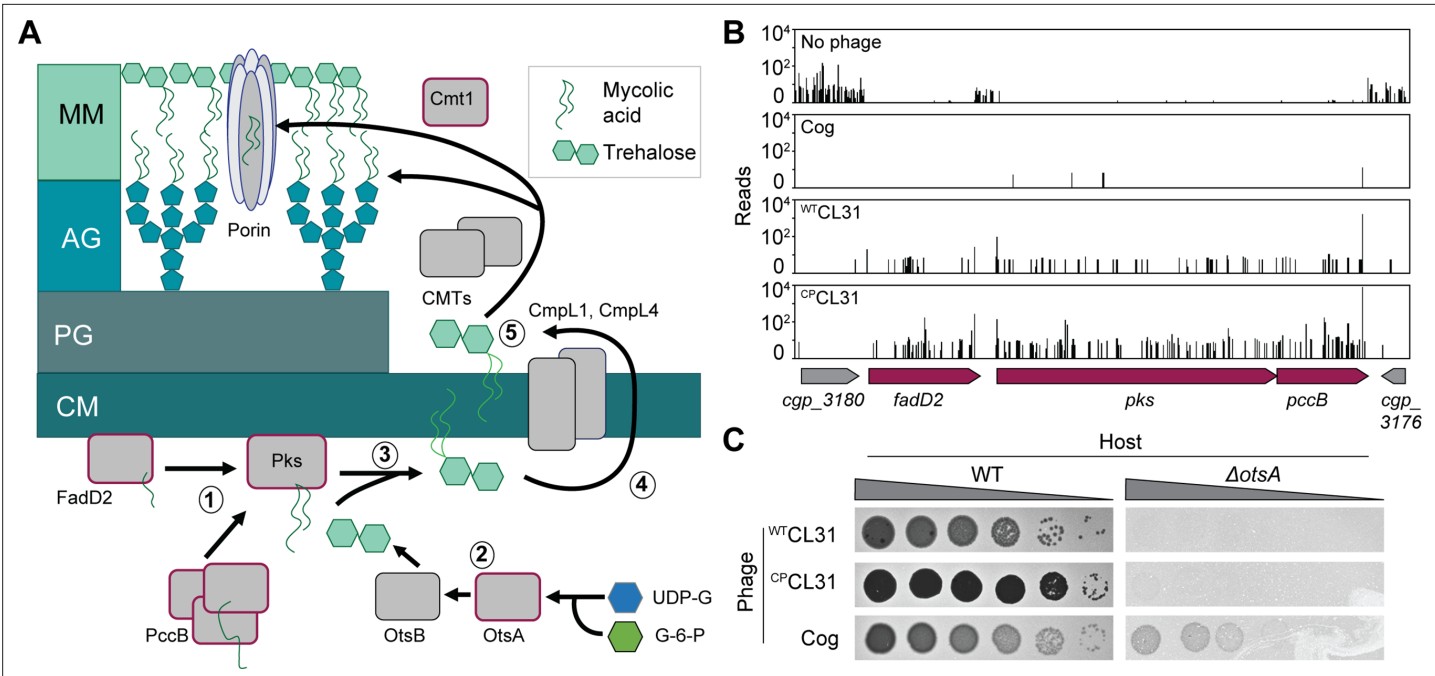

**Figure 5.** Mycolic acids are required for CL31 infection. (**A**) Cartoon of the mycolic acid synthesis pathway. Proteins outlined in burgundy were enriched for transposon insertions following CL31 challenge. (1) Long-chain fatty acids are synthesized by FadD2 and the acyl-CoA carboxylase complex (ACC), including PccB, for condensation by Pks. Concurrently, (2) OtsAB synthesizes trehalose from UDP-glucose (UDP-G) and glucose-6-phosphate (G-6-P). (3) The mycolic acid is attached to trehalose to form trehalose monomycolate (TMM) that is acetylated and reduced before (4) being transported across the membrane by the transporters CmpL1 and CmpL4. Once in the periplasm, (5) TMM is then used as a substrate of the corynemycoloyltransferases (CMTs), including Cmt1, to transfer the mycolate moieties to the arabinogalactan or proteins. (**B**) Transposon sequencing (Tn-Seq) insertion profile for a locus encoding core mycolic acid synthesis genes. (**C**) Validation of the Tn-Seq results for *ostA*. Ten-fold serial dilutions of lysates for the indicated phages were prepared and 3 µL of each dilution were spotted on a lawn of the indicated host strain.

The online version of this article includes the following figure supplement(s) for figure 5:

**Figure supplement 1.** CL31 phage requires genes involved in mycolic acid to infect *Corynebacterium glutamicum* (*Cglu*).

is then reduced and acetylated before being transported across the inner membrane by the CmpL1 or CmpL4 transporter (*Lea-Smith et al., 2007*; *Yamaryo-Botte et al., 2015*; *Yang et al., 2014*). Once exposed to the extracytoplasmic space, the TMM molecule is deacetylated by an unknown mechanism and either attached to the AG layer, transferred to proteins, or delivered to the mycomembrane. Covalent attachment of mycolic acids to proteins or the AG layer is performed by corynemycoloyltransferase (Cmt) enzymes (*Brand et al., 2003*; *Dautin et al., 2017*).

Unlike in mycobacteria, mycolic acids are not essential in *Cglu*. However, disruption of mycolic acid synthesis in *Cglu* still results in a severe growth defect. This growth defect is reflected in the low level of transposon insertions detected in loci-encoding factors involved in the biosynthetic pathway when the Tn-Seq profile of the untreated library was analyzed (*Figure 5B*). By contrast, following CL31 exposure, insertions in genes like *fadD2*, *pks*, and *otsA* were highly enriched despite their poor growth (*Figures 2 and 5B*, *Figure 5—figure supplement 1*). Insertions in these genes were also enriched upon Cog challenge, but to a lesser degree (*Figure 5B*, *Figure 5—figure supplement 1*), indicating that the Cog phage is less dependent on mycolic acids than CL31 for successful infection. To validate the Tn-Seq results, we tested for CL31 and Cog plaque formation on hosts that are defective for synthesizing mycolic acid. A *Δpks* host was unable to support phage infection by either phage, but the poor growth of the mutant made this result hard to interpret. On the other hand, inactivation of *otsA*, which is necessary for efficient trehalose synthesis and causes a less severe growth phenotype than the *Δpks* allele, greatly diminished the ability of both ^WTCL31 and ^CPCL31 to form plaques whereas plaque formation of Cog phage was much less adversely affected (*Figure 5C*). Thus, the deletion analysis confirms the more stringent requirement of CL31 phage for mycolic acids relative to Cog.

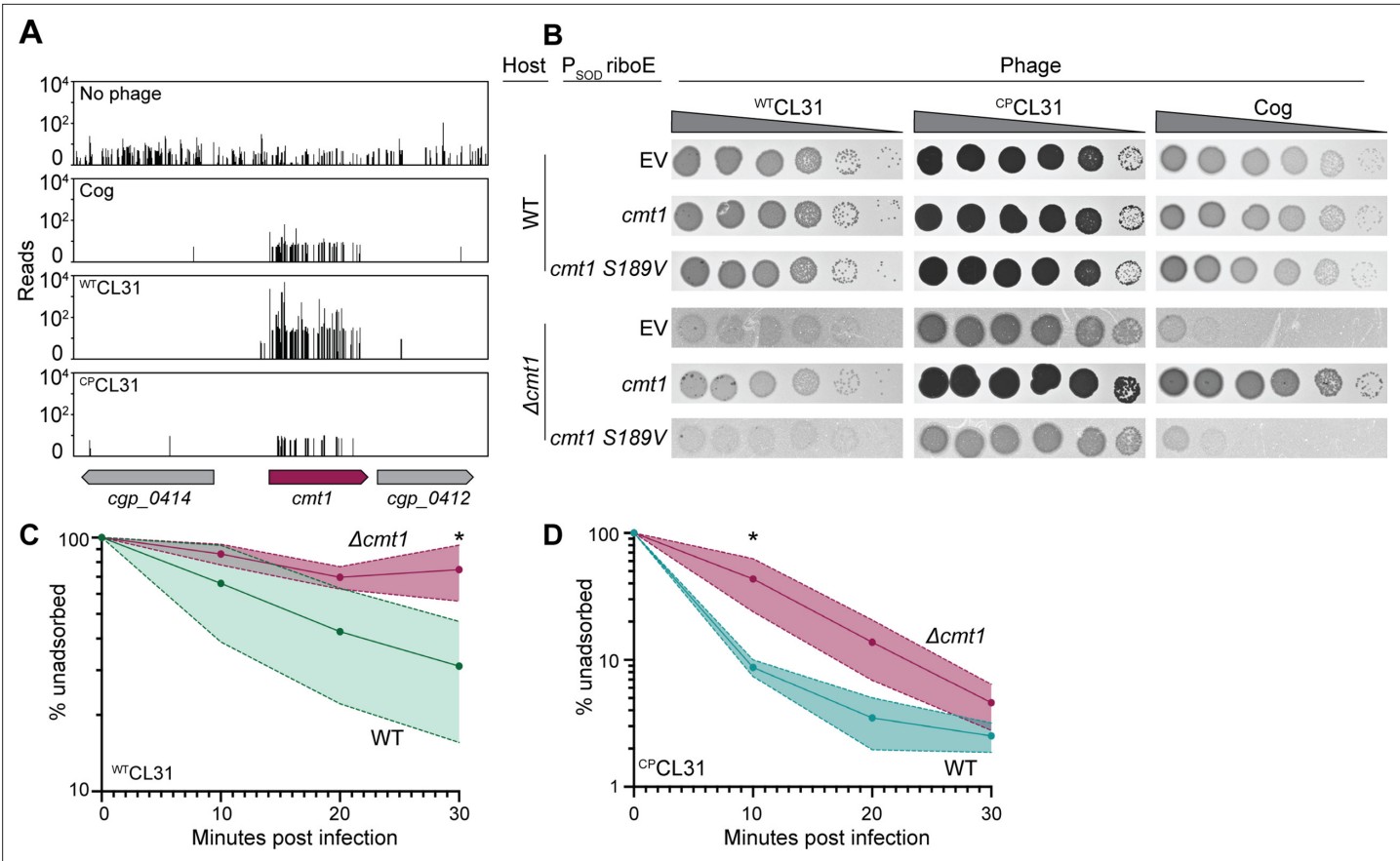

**Figure 6.** CL31 likely requires a post-translationally modified protein for efficient infection. (**A**) Transposon sequencing (Tn-Seq) insertion profiles of *cmt1* following phage challenge. (**B**) Ten-fold serial dilutions of lysates for the indicated phages were prepared and 3 µL of each dilution were spotted on a lawn of the indicated host strain. The strains contain an empty vector (EV) or the indicated gene expressed from the P$_{SOD}$ promoter with translation controlled by the *riboE* riboswitch. Protein production was induced with 1 mM theophylline. (**C–D**) Adsorption assay. The fraction of unadsorbed $^{WT}$CL31 (**C**) or $^{CP}$CL31 (**D**) phages was determined at the indicated timepoints following incubation with wild-type (WT) and *Δcmt1* cells of *Cglu*. Solid line indicates the mean and the shaded regions indicate the standard deviation from three independent experiments. Significance was determined by two-tailed t test. *$p < 0.05$.

The online version of this article includes the following source data for figure 6:

**Source data 1.** Adsorption data.

In addition to factors involved in mycolic acid synthesis, insertions in the *cmt1* gene encoding a corynemycoloyltransferase were also enriched following challenge with $^{WT}$CL31 (**Figures 2 and 6A**). Cmt1 (also referred to as MytC) is the only transferase in *Cglu* known to perform the *O*-mycoloylation post-translational modification involving the transfer of mycolic acid onto proteins (**Figure 5A**; *Huc et al., 2010*). Although the role of *O*-mycoloylation is poorly understood, it is thought to help localize proteins to the mycomembrane and stabilize their association with the envelope (*Carel et al., 2017*). Consistent with the Tn-Seq results, $^{WT}$CL31 is diminished in its ability to infect a *Δcmt1* mutant, forming hazy, diffuse plaques indicative of poor adsorption or injection (**Figure 6B**). Notably, $^{CP}$CL31 appeared to be much less dependent on Cmt1 for infection (**Figures 2B and 6A–6B**). However, its ability to infect the *Δcmt1* mutant was also reduced relative to wild-type cells. Instead of clear plaques, it formed turbid plaques resembling those of $^{WT}$CL31 on wild-type cells, indicating that the mutant phage still requires Cmt1 for optimal infectivity. Both ProtX and the PorAH porin are mycoloylated, and PorAB requires the *O*-mycoloylation modification in order to hetero-oligomerize and persist in the mycomembrane (*Huc et al., 2010*; *Carel et al., 2017*). We therefore tested the ability of Cog to plaque on the *Δcmt1* mutant. Even though *cmt1* was not a strong hit in the Tn-Seq analysis for Cog, the phage was also unable to efficiently infect *Δcmt1* cells (**Figure 6B**) as expected based on its requirement for ProtX and PorAH to efficiently infect *Cglu*. To investigate whether the activity of Cmt1

is required for phage infection, we tested the ability of a catalytic mutant of Cmt1 to complement the *Δcmt1* mutant for plaque formation (*Huc et al., 2013*). Unlike the wild-type enzyme, the inactivated Cmt1 was unable to restore normal plaque formation on the *Δcmt1* mutant for any of the phages (*Figure 6B*). Thus, it is *O*-mycoloylation activity of Cmt1 that is required for phage infection rather than the Cmt1 protein serving as the phage receptor itself.

To investigate the cause of the plaquing defect of cells lacking Cmt1, the ability of the CL31 phages to adsorb to mutant cells was assessed. The relatively poor adsorption of $^{WT}$CL31 to wild-type *Cglu* was made even worse by the inactivation of Cmt1, with the percentage of adsorbed phages after 30 min being greatly reduced for *Cglu Δcmt1* cells (*Figure 6C*). Notably, the percentage of adsorbed $^{CP}$CL31 phages after 30 min was not observed to be significantly different between infections of wild-type and *Δcmt1* cells (*Figure 6D*). However, the rate of adsorption of $^{CP}$CL31 was reduced for *Cglu Δcmt1* relative to wild-type cells. Notably, this reduced rate of adsorption is comparable to that observed for $^{WT}$CL31 on wild-type hosts and corresponds to a change in plaque morphology of $^{CP}$CL31 from clear to turbid. This result further supports the correlation between phage adsorption and plague turbidity and the hypothesis that the changes in Clg55 of CL31 enhance the affinity of phage for its receptor. Overall, the results thus far indicate that both CL31 and Cog require an *O*-mycoloylated protein(s) in the mycomembrane to efficiently infect *Cglu*, with Cog requiring ProtX and porins and CL31 utilizing one or more modified factors that remain to be identified.

## Identification of new gene involved in mycolic acid synthesis

To identify additional factors required for CL31 infection, we isolated spontaneous $^{WT}$CL31-resistant *Cglu* mutants in addition to the Tn-Seq analysis. *Cglu* cells were challenged with a high MOI (MOI = 5) of $^{WT}$CL31 and several isolates that survived the encounter were analyzed by whole-genome sequencing. As with the Tn-Seq analysis, the majority of the isolates had mutations in genes required for mycolic acid synthesis (*Figure 7—source data 1*), such as *fadD2, pccB,* and *otsA* or genes known to be associated with defects in mycolic acid synthesis, such as *deoC* (*de et al., 2020*). These findings were consistent with a recent report in which *pks* and *pccB* mutations were also found to promote CL31 resistance (*Hünnefeld et al., 2021*). In addition to mutations in well-characterized genes, we also isolated phage-resistant alleles in a gene of unknown function, *cgp_0475*, including a five-codon duplication (+EVLPL) within the reading frame. The *cgp_0475* gene was also a strong hit in the Tn-Seq analysis for both $^{WT}$CL31 and $^{CP}$CL31 infection (*Figure 2*). Cgp_0475 consists of a DUF2505 domain that is found almost exclusively in actinobacteria and falls within the wider class of the SRPBCC (START/RHO_alpha_C/PITP/Bet_v1/CoxG/CalC) superfamily, which includes lipid transfer proteins and polyketide cyclases (*Iyer et al., 2001*). Homologs of *cgp_0475* are found in both *Mtb* and *Mycobacterium smegmatis* (*Msmeg*), with the gene being duplicated in tandem in the latter (*Figure 7— figure supplement 1*). Structural models of Cgp_0475 and the *Mtb* homolog predict that it forms a helix-grip fold with a deep hydrophobic ligand binding pocket, a structure that has been implicated in lipid binding (*Iyer et al., 2001*). Thus, we named the gene <u>a</u>ctinobacterial <u>h</u>elix-grip <u>f</u>old A (AhfA). Similar to *pks* and other genes required for mycolic acid synthesis, Tn-Seq profiles of unchallenged cells showed few insertions in *ahfA* (*Figure 7A*), suggesting that there is a major fitness cost to inactivating this gene. This phenotype along with the high enrichment of hits in the mycolic acid synthesis pathway in the CL31 Tn-Seq analysis led us to investigate whether *ahfA* might represent a new gene required for mycolic acid synthesis.

We first validated the role of *ahfA* in facilitating phage infection by creating an in-frame deletion that minimized polar effects on the neighboring essential gene, *murB2*. As expected from the Tn-Seq analysis, both $^{WT}$CL31 and $^{CP}$CL31 were unable to form plaques on a *ΔahfA* mutant due in part to a reduced ability to adsorb to the cells (*Figure 7B and C*). Similar to other mutants inactivated for mycolic acid synthesis, the *Cglu ΔahfA* mutants had severe growth defects, forming small, rough colonies on plates and forming aggregates in liquid (*Figure 7D*). This defect was not due to polar effects of the *ahfA* deletion on the adjacent *murB2* gene as ectopic expression of *ahfA* was able to complement the phage plating defect and the slow growth phenotypes of the deletion mutant (*Figure 7B and D*). Additionally, the phage infection defect was not simply due to poor growth, as Cog was able to form plaques on the *ΔahfA* mutant, although at a slightly reduced efficiency (*Figure 7B*).

To investigate whether AhfA has a role in mycolic acid synthesis, we made use of the fluorescent trehalose analog 6-TAMRA-Trehalose (6-TMR-Tre), which is extracellularly incorporated by the Cmt

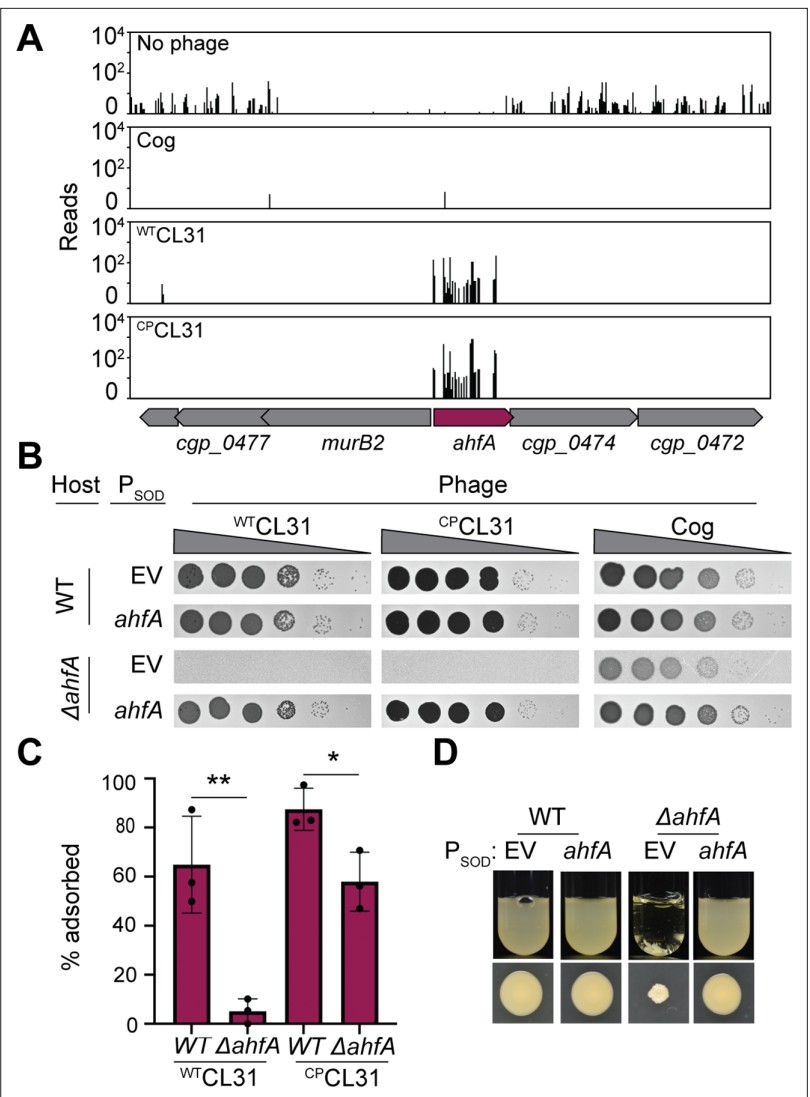

**Figure 7.** An uncharacterized gene, <u>a</u>ctinobacterial <u>h</u>elix-grip <u>f</u>old A (ahfA) (*cgp_0475*), is required for CL31 adsorption. (**A**) Transposon sequencing (Tn-Seq) insertion profile for the *ahfA* (*cgp_0475*) locus. (**B**) Ten-fold serial dilutions of lysates for the indicated phages were prepared and 3 µL of each dilution were spotted on a lawn of the indicated host strain. (**C**) The fraction of the indicated CL31 variant adsorbed was determined following a 30 min incubation with the indicated host strain. Dots indicate individual data points, bar height indicates mean, and error bars indicate standard deviation of three independent experiments. Significance was determined by two-tailed t test. *p<0.05, **p<0.005. (**D**) Representative images of overnight cultures and single colonies of the indicated strains after 3 days incubation at 30°C. Note the clumping and small colony phenotype of cells lacking *ahfA*.The strains in panels B and D contained an empty vector (EV) or the indicated gene constitutively expressed from the P_{SOD} promoter.

The online version of this article includes the following source data and figure supplement(s) for figure 7:

**Source data 1.** Single nucleotide polymorphisms (SNPs) in *Corynebacterium glutamicum* (*Cglu*) following challenge with <sup>WT</sup>CL31.

**Source data 2.** Adsorption data.

**Figure supplement 1.** <u>A</u>ctinobacterial <u>h</u>elix-grip <u>f</u>old A (AhfA) is required for efficient CL31 infection and has homologs in other mycolic acid synthesizing bacteria.

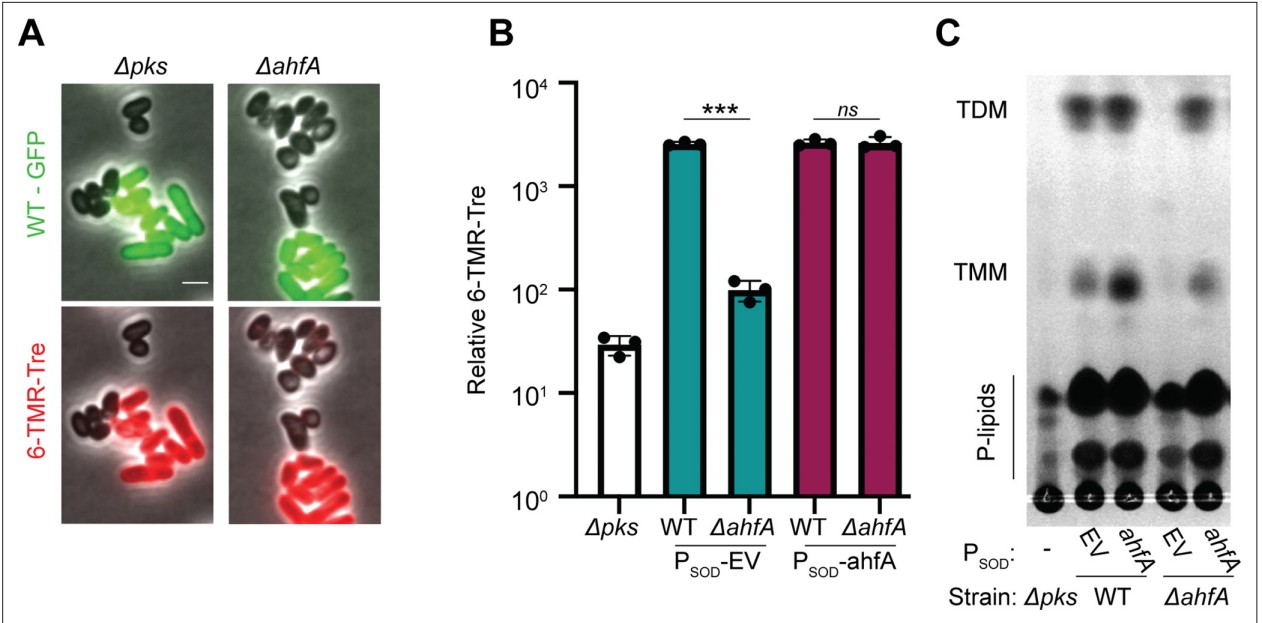

**Figure 8.** <u>A</u>ctinobacterial <u>h</u>elix-grip <u>f</u>old A (AhfA) is required for mycolic acid synthesis. (**A**) Wild-type (WT) cells expressing *gfp* or cells of the indicated strains lacking a fluorescent protein marker were grown to an $OD_{600}$ between 0.15 and 0.4 depending on strain fitness, stained with 6-TAMRA-Trehalose (6-TMR-Tre) for 30 min, washed, mixed together, and then applied to an agarose pad for visualization by fluorescence microscopy. Scale bar represents 2 µm. (**B**) Quantification of the extent of mycolic acid staining with 6-TMR-Tre as measured by a fluorescence plate reader. Dots indicate individual data points, bar height indicates the mean, and error bars indicate the standard deviation of three independent experiments. Significance was determined by two-tailed t test. ***p<0.001, *ns,* not significant. (**C**) TLC analysis of lipids extracted with chloroform/methanol from the indicated *Corynebacterium glutamicum* (*Cglu*) strains. The plate was developed in chloroform:methanol:water (30:8:1), dipped in primuline, and spots visualized with UV. Key species are labeled: TMM, trehalose monomycolate; TDM, trehalose dimycolate; P-lipids, phospholipids. The strains in panels B and C contained an empty vector (EV) or the indicated gene constitutively expressed from the $P_{SOD}$ promoter.

The online version of this article includes the following source data for figure 8:

**Source data 1.** Uncropped microscopy images.

**Source data 2.** 6-TAMRA-Trehalose (6-TMR-Tre) data.

**Source data 3.** TLC analysis of lipids extracted with chloroform/methanol from the indicated *Corynebacterium glutamicum* (*Cglu*) strains.

enzymes into newly synthesized and translocated mycolic acids in the mycomembrane (*Rodriguez-Rivera et al., 2018*). A mixture of wild-type cells that were constitutively expressing a green fluorescent protein (GFP) marker and GFP-free cells of *ΔahfA* or *Δpks* mutants were labeled with 6-TMR-Tre and visualized by fluorescence microscopy (*Figure 8A*). The GFP-expressing wild-type cells showed a strong peripheral labeling with 6-TMR-Tre as expected, whereas the *Δpks* mutant lacked any detectable signal. Label incorporation was also dramatically reduced in the *ΔahfA* mutant, but unlike the *Δpks* mutant, a very low but detectable signal was observed that was most evident at cell septa (*Figure 8A*). In addition to poor labeling with 6-TMR-Tre, cells of the two mutants appeared shorter and rounder than wild-type. A similar morphological phenotype has been reported for *Mtb* cells depleted for factors involved in mycolic acid synthesis (*de Wet et al., 2020*). To further investigate the 6-TMR-Tre labeling phenotype of cells inactivated for AhfA, dye incorporation was quantified using a fluorimeter. As expected, the *Δpks* mutant incorporated very little measurable dye (*Figure 8B*). Consistent with the imaging results, the *ΔahfA* mutant incorporated about threefold more dye than a mutant lacking the Pks synthase (*Figure 8A and B*), and this defect was complemented by expression of *ahfA* from a plasmid. The effect of AhfA inactivation on mycolic acid synthesis was additionally assessed by analyzing total lipid extracts by thin-layer chromatography. As suggested by the 6-TMR-Tre labeling assays, cells defective for AhfA showed a dramatic reduction in TMM and TDM lipids on par with a deletion in *pks* that could be restored with complementation (*Figure 8C*). Based on these results, we conclude that AhfA is a previously unidentified component of the mycolic acid synthesis pathway in *Cglu* and potentially other members of the Corynebacteriales.

## Discussion

Bacteriophages and their interactions with host cell receptors have long served as powerful tools for probing and monitoring bacterial surface assembly. For example, coliphages T4 and lambda played critical roles in defining the processes of LPS biogenesis and outer membrane protein transport, respectively. Additionally, phages of Gram-positive bacteria like *Bacillus subtilis* phage ΦT3 have served as useful probes for the synthesis of teichoic acids and other surface polymers (*Dunne et al., 2018*; *Freymond et al., 2006*). The receptors of phages that infect bacteria in the Corynebacteriales order have only just begun to be characterized. Gaining a greater understanding of the receptors recognized by these phages will aid the development of new tools for dissecting the assembly mechanism of the complex mycolata envelope surrounding their hosts. Among the multitude of phages that infect Corynebacteriales that have been isolated and genetically defined, receptor requirements for infection remain largely uncharacterized. In previous studies, lipids have been implicated in the initiation of infection by mycobacteriophages (*Imaeda and Blas, 1969*), but in only one example has a requirement for a defined surface glycolipid (*Chen et al., 2009*) been demonstrated. We therefore sought to globally define the factors required for phages to infect members of the Corynebacteriales using *Cglu* as a model system and two unrelated corynephages adapted to the lab strain MB001. Recent work has demonstrated the ability of high-throughput genome-wide analyses to reveal phage receptors for both characterized and novel phages that infect Gram-negative hosts (*Kortright et al., 2020*; *Mutalik et al., 2020*; *Adler et al., 2021*). Here, Tn-Seq analysis of resistant host mutants following phage challenge of a high-density transposon library successfully identified phage infection requirements, including both glycolipids and porin proteins in the mycomembrane. Additionally, the analysis identified new genes involved in AG modification and mycolic acid synthesis, highlighting the utility of the phage resistance profiling approach in revealing previously hidden aspects of mycolata envelope assembly for further functional characterization.

## Potential role of mycolic acids and/or *O*-mycoloylated proteins as phage receptors

The analysis of transposon insertion enrichment following challenge with CL31 or Cog revealed a requirement for genes in the mycolic acid synthesis pathway for optimal infection. The dependence on this gene set was near absolute for CL31 whereas the Cog phage was found to retain the ability to plaque on cells defective for mycolic acid synthesis, albeit at a reduced efficiency. Notably, the severity of the plating phenotypes was reversed for a mutant defective in the mycoloyltransferase Cmt1 with Cog phage showing the strongest defect and CL31, especially the [CP]CL31 variant, retaining plaquing ability on the Δ*cmt1* strain. Thus, the two phages appear to rely on mycolic acids for infection, but in differing ways.

For Cog, the dependence on Cmt1 for infection suggests that it uses a mycoloylated protein as a receptor. Accordingly, we observed a strong dependence on the mycomembrane proteins ProtX and PorA for Cog infection. 'Protein X' (ProtX) was initially identified through analysis of mycoloylated proteins with the *protX* gene being identified only recently (*Huc et al., 2010*; *Issa et al., 2017*). While little is known about the function of ProtX, the strong inhibition of plaque formation in parallel with the defects in adsorption indicate that ProtX is likely to be the outward facing receptor with which Cog engages to initiate infection. PorA and the other known porins are mycoloylated, and in several cases this modification has previously been found to depend on the Cmt1 transferase (*Carel et al., 2017*; *Huc et al., 2013*). Biochemical data has suggested that PorA forms an oligomer and can promote cation exchange across a lipid bilayer either alone or in complex with PorH (*Lichtinger et al., 2001*; *Barth et al., 2010*), and PorA is thought to be the most dominant protein in the mycomembrane (*Marchand et al., 2012*). Cog only required PorA for infection, not PorH, suggesting that PorA is functional in cells without PorH. We also observed that Cog plaquing was independent of the other porins PorB and PorC. Notably, Cog remained capable of efficiently adsorbing to Δ*porAH* cells but was defective in forming centers of infection following adsorption. Thus, Cog displays a relatively specific dependence on PorA for a stage of infection following its initial association with the cell surface, likely via ProtX. The finding that Cog has a stricter requirement for a mycoloylated protein for infection than the mycolic acid synthesis pathway itself suggests that a defect in mycolic acid synthesis and loss of the mycomembrane bypasses the need for ProtX and PorA to promote infection. We therefore infer that, following engagement with ProtX, Cog likely uses the porin as a conduit to penetrate the

mycomembrane for DNA injection and potentially engages with a secondary receptor in the AG layer to promote the injection process (see below).

The requirement of Cmt1 for CL31 infection is much less pronounced than that observed for Cog. The wild-type phage forms light hazy plaques on a lawn of the Δ*cmt1* mutant, but the absolute efficiency of plaque formation appears near normal (*Figure 6B*). Relative to the wild-type phage, the ᶜᴾCL31 mutant forms much more robust plaques on the Δ*cmt1* strain, but they are also hazier than the plaques formed on wild-type *Cglu*, which in the case of ᶜᴾCL31 are clear. The hazy plaque phenotype and the observed adsorption defect of cells lacking Cmt1 suggest that these phages have a reduced ability to bind to the cell surface of the Δ*cmt1* mutant, possibly due to the reduced display of a mycoloylated protein receptor. ᶜᴾCL31 may be less affected by the reduction in receptor concentration because the changes in its putative receptor binding protein Clg55 enhance its affinity for the receptor. Deletion analysis indicates that none of the known porins are used as the receptor. Also, none of the genes identified from the Tn-Seq as being required for CL31 infection besides *cmt1* encode a known mycoloylated protein (*Kavunja et al., 2016*). Therefore, it is possible that CL31 employs multiple redundant protein receptors, which could explain why a strong phage resistance phenotype requires inactivation of the mycolic acid synthesis pathway. However, this observation could also indicate that CL31 directly recognizes glycolipids in the mycomembrane as part of the infection process instead of, or in addition to, mycoloylated protein(s). Although further work will be required to test these possibilities and determine the identity of the putative receptor(s), the Tn-Seq analysis clearly implicates mycolic acids in *Cglu* infection by corynephages.

## AG modification and Cog infection

In addition to the PorA and Cmt1 requirements, we also observed that the Cog phage was dependent on a collection of genes implicated in AG modification to properly infect *Cglu*. As with mutants lacking PorA, Cog was found to effectively adsorb to *Cglu* cells lacking these genes. Thus, the putative AG modification is needed for a stage of infection after engagement with the *Cglu* cell surface, which is consistent with the fact that the AG layer is covered by the mycomembrane. A requirement for modified cell wall polymer is common among phages that infect Gram-positive bacteria (*Dunne et al., 2018*). For example, phages that require specific modifications of wall teichoic acids (WTAs) or lipoteichoic acids are often used to type strains, such as the use of phages that infect *Listeria monocytogenes* to differentiate pathogenic serovars (*Loessner and Busse, 1990*). Additionally, structural studies of the *Staphylococcus aureus* phage 80α has led to the model that binding to WTAs leads to conformational changes in the virion that allow for activation of a structurally encoded phage hydrolase to degrade the PG layer (*Kizziah et al., 2020*). We therefore suspect that AG modification and association with the AG layer may be playing a similar role for Cog in promoting *Cglu* infection.

The high degree of similarity between the *ppgS*, *cgp_0658, cgp_0391*, and *cgp_0393* genes with the analogous gene set in *Mtb* strongly suggests that they encode a related AG modification system. Whether this system also adds GalN moieties to the AG as in *Mtb* or modifies the envelope layer with a different decoration remains unclear. Notably, there does not appear to be an analog of *cgp_0396* in the mycobacterial system. However, given the similar phenotypic profile of *cgp_0396* with the AG modification gene set and its importance for Cog infection, we conclude that this gene is also likely to encode an AG modification enzyme. Given that Cgp_0396 is part of the GT2 family of glycosyltransferases, we propose that it adds an additional sugar moiety to the PolyP-P-linked GalN synthesized by the other components of the system, but this possibility will require further investigation.

## A new factor involved in mycolic acids synthesis

In addition to genes known to be important for mycolic acid synthesis, resistance profiling with the CL31 phage also identified AhfA (Cgp_0475) as a new factor required for the biogenesis of this critical envelope component. Unlike mutants lacking the Pks enzyme that directly synthesizes the lipid chain of mycolic acids, in which the TMM and TDM lipids are undetectable (*Portevin et al., 2004*), cells inactivated for AhfA retain a very low but detectable level of these molecules (*Figure 8B–C*). Thus, although AhfA is critical for proper mycolic acid synthesis, it does not appear to be absolutely required for TMM production. AhfA is predicted to have a helix-grip fold domain implicated in lipid binding. The protein also lacks a predicted secretion signal, suggesting it is cytoplasmic. Notably, mutants of *Cglu* inactivated for its redundant pair of transmembrane TMM transporters (flippases) CmpL1 and

CmpL4 also display a loss of extractable TMM glycolipids, indicating that proper glycolipid transport is somehow linked to its synthesis (*Yang et al., 2014*). We therefore propose that AhfA may similarly function in TMM transport, potentially as a carrier protein that shuttles the lipids from the synthetic enzymes to the CmpL1 and CmpL4 transporters.

AhfA is conserved in both *Msmeg* and *Mtb*, but it has not been found to be essential in these organisms that require mycolic acid synthesis for viability. *Msmeg* possesses two *ahfA*-like genes in tandem (MSMEG_0927 and MSMEG_0926) at a similar locus to that of *ahfA* in *Cglu*. *Mtb*, on the other hand, only encodes a single *ahfA* gene (Rv0481c) at this genomic position, but both *Mtb* and *Msmeg* encode other predicted SRPBCC family proteins, at least one of which is essential (MSMEG_0129/Rv0164) (*Zheng et al., 2018*). Thus, the closest homolog of *Cglu ahfA* in these organisms may not be essential due to redundancy with other related genes. Nevertheless, the strong requirement for AhfA for mycolic acid synthesis in *Cglu* indicates that this gene and/or its relatives are likely to play important roles in the biogenesis of the mycomembrane and further dissection of their function promises to reveal new ways of compromising the pathway for antibiotic development.

# Methods

## Key resources table

| Reagent type (species) or resource | Designation | Source or reference | Identifiers | Additional information |
|---|---|---|---|---|
| Strain, strain background (*Echerichia coli*) | *F– hsdR17 deoR recA1 endA1 phoA supE44 thi-1 gyrA96 relA1 Δ(lacZYA-argF) U169 Φ80dlacZΔM15 ****add pir* | Gibco BRL | DH5α | |
| Strain, strain background (*Cglu*, MB001) | MB001 (ATCC 13032 ΔCGP1 (cg1507-gp1524) ΔCGP2 (cg1746-1752) ΔCGP3 (cg1890-cg2071)) | *Baumgart et al., 2013* | H60 | |
| Strain, strain background (*Cglu*) | Environmental *Cglu* isolate ATCC 15990 | Félix d'Hérelle Reference Center for Bacterial Viruses | ATCC 15990 | |
| Strain, strain background (*Cglu*) | Environmental *Cglu* isolate LP-6 | Félix d'Hérelle Reference Center for Bacterial Viruses | LP-6 | |
| Strain, strain background (*Cglu*, MB001) | ΔRM (Δcgp_0844) | This paper | H722 | Bernhardt Lab |
| Strain, strain background (*Cglu*, MB001) | ΔppgS (Δcgp_0394) | This paper | H2258 | Bernhardt Lab |
| Strain, strain background (*Cglu, MB001*) | Δcgp_0658 | This paper | H2259 | Bernhardt Lab |
| Strain, strain background (*Cglu*, MB001) | Δcgp_0396 | This paper | H2260 | Bernhardt Lab |
| Strain, strain background (*Cglu*, MB001) | ΔprotX (Δcgp_2875) | This paper | H1111 | Bernhardt Lab |
| Strain, strain background (*Cglu*, MB001) | ΔporBC (Δcgp_1108c) | This paper | H1239 | Bernhardt Lab |
| Strain, strain background (*Cglu*, MB001) | ΔporAH (Δcgp_3008–3009) | This paper | H1152 | Bernhardt Lab |
| Strain, strain background (*Cglu*, MB001) | ΔporH (Δcgp_3009) | This paper | H1241 | Bernhardt Lab |
| Strain, strain background (*Cglu*, MB001) | ΔporA (Δcgp_3008) | This paper | H1247 | Bernhardt Lab |
| Strain, strain background (*Cglu*, MB001) | ΔotsA (Δcgp_2907) | This paper | H1666 | Bernhardt Lab |
| Strain, strain background (*Cglu*, MB001) | Δpks (Δcgp_3178) | This paper | H2261 | Bernhardt Lab |
| Strain, strain background (*Cglu*, MB001) | Δcmt1 (Δcgp_0413) | This paper | H1664 | Bernhardt Lab |

*Continued on next page*

*Continued*

| Reagent type (species) or resource | Designation | Source or reference | Identifiers | Additional information |
|---|---|---|---|---|
| Strain, strain background (*Cglu*, MB001) | Δ*ahfA* (Δ*cgp_0475*) | This paper | H2262 | Bernhardt Lab |
| Strain, strain background (*Cglu*, MB001) | WT P$_{SOD}$-*ahfA* | This paper | H2272 | pK-PIM derivative encoding P$_{SOD}$-*ahfA* isothermally assembled and directly transformed into *Cglu*. |
| Strain, strain background (*Cglu*, MB001) | Δ*ahfA* P$_{SOD}$-*ahfA* | This paper | H2273 | pK-PIM derivative encoding P$_{SOD}$-*ahfA* isothermally assembled and directly transformed into *Cglu*. |
| Strain, strain background (Bacteriophage) | env CL31 (phage isolated on *Cglu* ATCC 15990) | Félix d'Hérelle Reference Center for Bacterial Viruses **Trautwetter et al., 1987** | HER#229 | |
| Strain, strain background (Bacteriophage) | env Cog (phage isolated on *Cglu* LP-6) | Félix d'Hérelle Reference Center for Bacterial Viruses (**Sonnen et al., 1990**) | HER#360 | |
| Strain, strain background (Bacteriophage) | WT CL31 (*T19334G*) | This paper | ACMΦ2 | Bernhardt Lab |
| Strain, strain background (Bacteriophage) | CP CL31 (*T19334G, T24296A*) | This paper | ACMΦ9 | Bernhardt Lab |
| Strain, strain background (Bacteriophage) | Cog (*A19202T*) | This paper | ACMΦ8 | Bernhardt Lab |
| Recombinant DNA reagent (plasmid) | P$_{SOD}$ riboE EV, inducible (Kan$^R$, pK-PIM derivative encoding P$_{SOD}$-riboE1 empty vector) | This paper | pACM185 | Bernhardt Lab, |
| Recombinant DNA reagent (plasmid) | P$_{SOD}$ riboE *cmt1*, inducible (Kan$^R$, pK-PIM derivative encoding P$_{SOD}$-riboE1-*cmt1*) | This paper | pACM64 | Bernhartdt Lab, |
| Recombinant DNA reagent (plasmid) | Kan$^R$, pK-PIM derivative encoding P$_{SOD}$-riboE1-*cmt1 S189V* (P$_{SOD}$ riboE *cmt1 S189V*, inducible) | This paper | pACM66 | Bernhardt Lab |
| Recombinant DNA reagent (plasmid) | P$_{SOD}$ EV, constitutive (Kan$^R$, pK-PIM derivative encoding P$_{SOD}$-empty vector) | This paper | pEMH7 | Bernhardt Lab |
| Recombinant DNA reagent (plasmid) | P$_{TAC}$-*gfp*, inducible (Kan$^R$, P$_{TAC}$-eGFP, pGA1 mini replicon) | **Ravasi et al., 2012** | pTGR5 | |
| Recombinant DNA reagent (plasmid) | Kan$^R$, pCRD206 derivative containing an insert covering upstream and downstream of *cgp_0844* | This paper | pJWS51 | Bernhardt Lab |
| Recombinant DNA reagent (plasmid) | Kan$^R$, pCRD206 derivative containing an insert covering upstream and downstream of *cgp_0394* (*ppgS*) | This paper | pACM285 | Bernhardt Lab |
| Recombinant DNA reagent (plasmid) | Kan$^R$, pCRD206 derivative containing an insert covering upstream and downstream of *cgp_0658* | This paper | pACM286 | Bernhardt Lab |
| Recombinant DNA reagent (plasmid) | Kan$^R$, pCRD206 derivative containing an insert covering upstream and downstream of *cgp_0396* | This paper | pACM284 | Bernhardt Lab |
| Recombinant DNA reagent (plasmid) | Kan$^R$, pCRD206 derivative containing an insert covering upstream and downstream of *cgp_2875* (*protX*) | This paper | pEMH1 | Bernhardt Lab |
| Recombinant DNA reagent (plasmid) | Kan$^R$, pCRD206 derivative containing an insert covering upstream and downstream of *cgp_1108-cgp_1109* (*porBC*) | This paper | pEMH52 | Bernhardt Lab |

*Continued on next page*

*Continued*

| Reagent type (species) or resource | Designation | Source or reference | Identifiers | Additional information |
|---|---|---|---|---|
| Recombinant DNA reagent (plasmid) | Kan$^R$, pCRD206 derivative containing an insert covering upstream and downstream of *cgp_3008–3009* (*porAH*) | This paper | pEMH10 | Bernhardt Lab |
| Recombinant DNA reagent (plasmid) | Kan$^R$, pCRD206 derivative containing an insert covering upstream and downstream of *cgp_3009* (*porH*) | This paper | pEMH9 | Bernhardt Lab |
| Recombinant DNA reagent (plasmid) | Kan$^R$, pCRD206 derivative containing an insert covering upstream and downstream of *cgp_3008* (*porA*) | This paper | pEMH16 | Bernhardt Lab |
| Recombinant DNA reagent (plasmid) | Kan$^R$, pCRD206 derivative containing an insert covering upstream and downstream of *cgp_2907* (*otsA*) | This paper | pACM30 | Bernhardt Lab |
| Recombinant DNA reagent (plasmid) | Kan$^R$, pCRD206 derivative containing an insert covering upstream and downstream of *cgp_3178* (*pks*) | This paper | pACM175 | Bernhardt Lab |
| Recombinant DNA reagent (plasmid) | Kan$^R$, pCRD206 derivative containing an insert covering upstream and downstream of *cgp_0413* (*cmt1*) | This paper | pACM20 | Bernhardt Lab |
| Recombinant DNA reagent (plasmid) | Kan$^R$, pCRD206 derivative containing an insert covering upstream and downstream of *cgp_0475* (*ahfA*) | This paper | pACM230 | Bernhardt Lab |
| Chemical compound. drug | 6-TAMRA-Trehalose (6-TMR-Tre) | Tocris | 6802 | |

## Bacterial growth conditions

*Cglu* strains were grown in BHI medium, except for *Cglu* mutant isolates with mycolic acid synthesis defects, which were grown in BHI supplemented with 9.1% sorbitol. Unless otherwise indicated, growth was with aeration at 30°C. Induction of ectopic constructs was performed with theophylline (theo) or isopropy β-D-1-thiogalactopyranoside (IPTG) as indicated. *E. coli* strains were grown in LB media with aeration at 37°C unless otherwise noted. When necessary, 15 or 25 µg/mL kanamycin (Kan) was added to *Cglu* or *E. coli* cultures, respectively.

## Strain construction and cloning

Plasmids were created through isothermal assembly (ITA) and were either transformed into *E. coli* or electroporated directly into *Cglu*. Competent *Cglu* cells were prepared as previously described (*Lim et al., 2019*). Chromosomal deletion mutations were generated using the pCRD206 temperature-sensitive plasmid (*Okibe et al., 2011*). Briefly, 500-1 kb fragments containing upstream and downstream homology bracketing the desired gene deletion were amplified and assembled into pCRD206 through ITA. The assembled plasmids were electroporated into *Cglu* and transformants were isolated on BHI-Kan at 25°C. Plasmid integrants were then isolated by streaking transformants on BHI-Kan followed by growth at 30°C. Integrants were then plated onto BHI supplemented with 10% sucrose to select against plasmid-encoded *sacB* for the isolation of deletion mutants. The desired deletion mutants were identified by diagnostic PCR using primers upstream and downstream of the integration site. Complementation was performed by a derivative of the integrating plasmid pK-PIM (*Oram et al., 2007*). Integration in *Cglu* was validated by colony PCR. Primers used to construct strains and plasmids are listed in *Supplementary file 1*.

## Phage growth and plaquing

Phage spot titer assays were performed by first adding 200 µL of an overnight *Cglu* culture to 4 mL of molten BHI top agar (0.5% agar). The mixture was then vortexed and applied to a standard BHI agar plate (1.5% agar). After allowing the plate to dry under a flame for 10 min, 3 µL of each dilution in a 10-fold dilution series of phage were applied to the top agar and allowed to dry on the surface. Dried plates were then incubated overnight at 30°C and imaged using a digital scanner. To generate high-titer phage stocks, 200 µL of a mid-log *Cglu* culture (or culture of 0.3<OD<0.5) was mixed with

10 µL of various dilutions of a phage stock. After incubation at room temperature for 10 min, 4 mL of molten BHI top agar was added and poured onto a Petri dish. Each mixture was then vortexed and applied to a separate standard BHI agar plate. After solidifying, plates were incubated at room temperature overnight. Plates displaying confluent lysis of the *Cglu* lawn were overlaid with 8 mL of STE buffer (100 mM NaCl, 10 mM Tris pH 8, 1 mM EDTA) and rocked overnight at 4°C. The overlay buffer was then collected and centrifuged to remove bacterial debris. The remaining supernatant was either concentrated through an Amicon centrifugal column (50 kDa cutoff) or precipitated with poly-ethylene glycol (PEG) 8000 and resuspended in SM buffer (100 mM NaCl, 8 mM $MgSO_4$, 50 mM Tris-Cl pH 7.5, 0.002% gelatin). For quantification of phage stock titers, *Cglu* hosts were grown to midlog ($OD_{600}$=0.3–0.4), and 100 µL of the culture was mixed with 10 µL of a phage stock dilution. Following adsorption (10 min), the mixture was then added to BHI top agar and plated as above for phage stock preparation. Following an overnight incubation at 30°C, plaques were counted to determine the phage titer of the stock in question.

## Adaptation of phage to lab *Cglu*

Phage acquired from the Félix d'Hérelle Reference Center for Bacterial Viruses were plaque purified three times on their respective environmental hosts and stocked in SM buffer ([env]CL31 and [env]Cog, respectively). To adapt CL31, diluted purified [env]CL31 was plaqued on *Cglu* MB001 in BHI top agar and a single plaque was recovered into SM buffer. Phage from this plaque was again plated onto MB001, wherein two plaque morphologies were apparent. Each plaque morphotype ([WT]CL31 and [CP]CL31) was picked and plaque purified independently two more times. Further isolation of other [CP]CL31 alleles was performed by plaquing [WT]CL31 onto MB001, picking clear plaques formed among a high density of turbid plaques, and then plaque purifying on MB001. [env]Cog was initially unable to form plaques on MB001. We therefore tested the ability of [env]Cog to plaque on MB001 derivatives with mutations in potential restriction-modification (RM) systems, as well as MB001 grown on different media (LB 0.4% glucose (LBG) compared to BHI; *Figure 1—figure supplement 1A* ). Clear plaques were formed on a lawn of the RM mutant grown in LBG top agar following infection with a high concentration of phage. These plaques were purified then plated on wild-type MB001 in BHI top agar, and the resulting clear plaques were purified three times for use in the phage challenge and other experiments.

## DNA sequencing of phages

Phage gDNA was purified from high titer stocks using the Qiagen DNeasy kit. Libraries for sequencing were prepped using a modified Nextera protocol as previously described (*Baym et al., 2015*) with some modifications (*Rohs et al., 2018*). The library was sequenced on a MiSeq (Illumina) using a MiSeq Reagent Kit v2 Nano with 150×150 paired end run. All NGS data from this study are deposited under BioProject PRJNA834153. Reads were processed and de novo assembled using the CLC Genomics Workbench software (Qiagen). Assembled genomes were annotated using the RAST annotation pipeline and GeneMarkS (*Aziz et al., 2008*; *Besemer et al., 2001*). Allele-specific sequencing of CL31 was achieved by PCR amplification of template acquired from boiled plaques and analyzed via Sanger sequencing.

## Analysis of phage adsorption and center of infection formation

For adsorption rates, overnight cultures of *Cglu* were back diluted (1:100) and grown to an $OD_{600}$~ 0.3. Phage was added to a 1 mL aliquot of the culture at an MOI between 0.1 and 0.01. The infection mixture was then vortexed and incubated at 30°C with rolling for the indicated timepoints. At each timepoint, 200 µL of the infected sample was removed and centrifuged for 3 min at 5000×*g*. The supernatant was collected and tittered in single or technical duplicate on wild-type *Cglu*. For measurements of terminal phage adsorption to mutant strains, overnight cultures were grown as above. Phage was added to a 200 µL aliquot of culture at a concentration of ~4 × $10^5$ pfu/mL (0.001<MOI<0.01), vortexed, and incubated with rotation at 30°C for 30 min. The same phage inoculum was added to 200 µL of sterile media, vortexed, and immediately harvested to determine phage input. The infection and mock-infection mixtures were centrifuged to separate cells from free phages, and the supernatant was collected and tittered in technical duplicate on wild-type *Cglu* (mid-log culture) to determine the percent of adsorbed phages. The remaining cell pellet was used to quantify the centers of infection. The pellet was washed twice in BHI supplemented with 50 mM sodium citrate to remove

any remaining unbound phages, resuspended in 100 µL BHI, and tittered in technical duplicate in a lawn of wild-type *Cglu* (mid-log culture). For improved visualization of Cog plaque formation, 10 mM $MgCl_2$ was added to the top agar. Each adsorption and infectious centers experiment was performed independently in biological triplicates.

## Phage infection and Tn-Seq

Two independent rounds of infection of transposon-mutagenized cells were performed on a previously created *Cglu* high-density transposon library (*Lim et al., 2019*). Frozen aliquots of the library were seeded into a BHI culture supplemented with Kan to an approximate $OD_{600}$ of 0.01. This culture was grown at 30°C to an $OD_{600}$=0.3–0.4, and 2 mL aliquots were removed into fresh culture tubes to be infected with phage at an MOI of 5 or to serve as an uninfected control. The culture tubes were then returned to the incubator and rolled for 30 min. Following incubation, the cultures were pelleted, washed once with BHI, and resuspended in 1 mL BHI. For cultures that had been exposed to phage, the entirety of the resuspension was plated as 100 µL aliquots across 10 BHI Kan plates. For the uninfected cultures, the resuspension was diluted to $10^{-2}$ in 1 mL BHI and similarly plated. Plates were incubated overnight at 30°C and followed by a second day of incubation at room temperature. After incubation, the colonies from each set of plates counted and pooled. gDNA was prepped with the Wizard Genomic DNA Purification Kit (Promega) and further cleaned using the Genomic DNA Clean & Concentrator (Zymo). Library preparation was performed as previously described (*Lim et al., 2019*). Due to predicted low complexity, phage-challenged libraries were pooled at a 1:100 ratio relative to the unchallenged libraries. Libraries were sequenced on a MiSeq with 1×50 single paired end reads. All NGS data from this study are deposited under BioProject PRJNA834153. Reads were then trimmed with trimmomatic 0.36 mapped to the *Cglu* MB001 genome (NCBI NC_022040.1) using bowtie 1.0.0 (*Bolger et al., 2014*; *Langmead et al., 2009*). To identify genes of interest in phage conditions, we normalized the number of phage infected reads to the total number of paired control reads and calculated the fold change in normalized read counts between each phage and its paired uninfected control. Additionally, a Mann-Whitney U test was used to determine whether the differences were statistically significant. Visual inspection of the insertion sites was performed using the Sanger Artemis Genome Browser and Annotation Tool. Genes of interested were determined to be those in which the fold change of insertions was greater than $log_2$ of 2 and with more than one unique insertion in both replicates of the Tn-seq. Additionally, an exception was made for Cog *porA* and *porH*, due to the small size of the genes.

## Bacterial spot plating for single-colony imaging

Bacteria were scraped from a plate and resuspended in BHI. The $OD_{600}$ of the resuspension was measured and normalized to $OD_{600}$=1. The suspension was then serially diluted 10-fold, and 5 µL of each dilution was spotted onto a plate. The plate was then incubated at 30°C for 3 days, and single colonies were imaged.

## Visualization of fluorescently labeled mycolic acids

Overnight cultures of *Cglu* were back-diluted to an $OD_{600}$=0.025, grown to an $OD_{600}$ between 0.15 and 0.5 depending on fitness of mutant, and normalized to 1 mL of $OD_{600}$=0.3. WT cells ectopically expressing $P_{TAC}$-GFP were grown with Kan and 0.1 mM IPTG. Cells were pelleted for 3 min at 5000×*g*, resuspended in 100 µL BHIS with 100 µM 6-TMR-Tre (Tocris), and incubated in the dark for 30 min. Cells were then re-pelleted, washed 2× with BHIS, and resuspended in BHIS. Wild-type and mutant cells were mixed together in a 1:5 ratio and spotted onto BHIS 1% agarose pads. Images were acquired on a Nikon Ti-E inverted widefield microscope with a motorized stage, a perfect focus system, and a 1.45 NA Plan Apo ×100 Ph3 DM objective lens with Cargille Type 37 immersion oil. Fluorescence imaging was performed with Lumencore SpectraX LED Illumination with images in the ET-GFP and ET-mChrery channels using Chroma 49002 and 49008 filter sets, respectively. Acquired images were taken with an Andor Zyla 4.2 Plus sCMOS camera (65 nm pixel size) with Nikon Elements acquisition software (v5.10). Images were rendered for publication with Fiji and contrast was normalized between the samples.

## Quantification of 6-TMR-Tre incorporation

Cells were grown, pelleted, and stained as for microscopy. After 30 min of staining, the pellets were washed 2× in PBS and resuspended in 1 mL PBS. An aliquot of each cell suspension (200 µL) was

added to a Corning black-welled, clear bottom 96-well plate in technical duplicate and read in a Tecan Infinite 200 Pro. Each sample had the $OD_{600}$ read, as well as the fluorescence with excitation at 532 nm and emission at 580 nm. Each $OD_{600}$ and fluorescence measurement was normalized to a PBS-only control, and the relative 6-TMR-Tre intensity was calculated by dividing the normalized fluorescence intensity by the normalized $OD_{600}$. Each value was measured independently in triplicate.

## Lipid extraction and analysis

Overnight cultures were diluted to an $OD_{600}$=0.025 and grown at 30°C to late log, about 6 hr. Cells were pelleted for 10 min at 3000×$g$ at 4°C, washed with cold PBS, and stored at –20°C until lipid extraction. The frozen pellets were resuspended in PBS and moved to a pre-weighed microcentrifuge tube, and the pellet weight was attained. For each sample 12 mg of cells were used per extraction. First, the pellet was resuspended in 1:2 chloroform ($CHCl_3$):methanol (MeOH) for 2 hr at room temperature, pelleted for 5 min at 500×$g$, and the organic layer containing the lipids was collected. The pellet was then sequentially extracted with 1:1 $CHCl_3$:MeOH, and 2:1 $CHCl_3$:MeOH, with the organic layers collected between each extraction. The pooled organics were then dried in an Eppendorf Vacufuge Plus speedvac and stored at –20°C. The dried organics were then cleaned by resuspending in 1:1 $CHCl_3$:$H_2O$, vortexing, and centrifuging to separate the phases. The organic layer was again collected and dried. The dried lipids were resuspended in $CHCl_3$ and spotted onto a glass-backed silica gel 60 TLC plate (MilliporeSigma) for analysis. The TLC was run in 30:8:1 $CHCl_3$:MeOH:$H_2O$, dipped into 0.01% primuline in 80% acetone, and visualized under UV on an Azure Biosystems c150 imager.

## Acknowledgements

The authors gratefully acknowledge all members of the Bernhardt and Rudner labs for advice and helpful discussions. In particular, we would like to thank Joel Sher for assistance with the Tn-Seq experiments and Elizabeth M Hart for creating and sharing the *protX* and porin-related mutants. We also thank the microscopy services provided by Paula Montero Llopis and her team at the Microscopy Resources on the North Quad (MicRoN) core facility at HMS. This work was supported by Investigator funds from the Howard Hughes Medical Institute. ACM was supported in part by the Life Sciences Research Foundation, where she is a Simons Fellow.

## Additional information

### Funding

| Funder | Grant reference number | Author |
|---|---|---|
| Howard Hughes Medical Institute | | Thomas G Bernhardt |
| Life Sciences Research Foundation | | Amelia C McKitterick |

The funders had no role in study design, data collection and interpretation, or the decision to submit the work for publication.

### Author contributions

Amelia C McKitterick, Thomas G Bernhardt, Conceptualization, Funding acquisition, Investigation, Methodology, Writing - original draft, Writing - review and editing

### Author ORCIDs

Amelia C McKitterick ⬤ http://orcid.org/0000-0001-7290-2781
Thomas G Bernhardt ⬤ http://orcid.org/0000-0003-3566-7756

### Decision letter and Author response

Decision letter https://doi.org/10.7554/eLife.79981.sa1
Author response https://doi.org/10.7554/eLife.79981.sa2

## Additional files

### Supplementary files
- Supplementary file 1. Primer list.
- MDAR checklist

### Data availability
Sequencing data generated from this study have been deposited in the NCBI Sequence Read Archive under BioProject PRJNA834153. All other data generated or analyzed during this study are provided in the manuscript and supporting files.

The following dataset was generated:

| Author(s) | Year | Dataset title | Dataset URL | Database and Identifier |
|---|---|---|---|---|
| McKitterick AC, Bernhardt TG | 2022 | Phage resistance profiling identifies new genes required for biogenesis and modification of the corynebacterial cell envelope | https://www.ncbi.nlm.nih.gov/sra/PRJNA834153 | NCBI Sequence Read Archive, PRJNA834153 |

The following previously published dataset was used:

| Author(s) | Year | Dataset title | Dataset URL | Database and Identifier |
|---|---|---|---|---|
| Sher JW, Lim HC, Bernhardt TG | 2020 | Phenotypic profiling of a Corynebacterium glutamicum transposon library | https://www.ncbi.nlm.nih.gov/bioproject/PRJNA610521 | NCBI BioProject, PRJNA610521 |

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
