## [Editor Report]

The authors perform a Transposon-Sequencing screen to determine bacterial factors (including receptors) important for infection by two phages in the model bacterium Corynebacterium glutamicum. Using their established high-density transposon library, they identify genes required for infection with the phages Cog and CL31. They also identified a spontaneous phage-resistant mutant that led to the discovery of a gene involved in mycolic acid synthesis. Overall, the work is of broad interest to scientists in the field of cell wall biogenesis, phage infection, and bacterial cell biology.

---

## [Decision Letter]

**Decision letter after peer review:**

Thank you for submitting your article "Phage resistance profiling identifies new genes required for biogenesis and modification of the corynebacterial cell envelope" for consideration by *eLife*. Your article has been reviewed by 3 peer reviewers, and the evaluation has been overseen by a Reviewing Editor and Bavesh Kana as the Senior Editor. The following individuals involved in the review of your submission have agreed to reveal their identity: William R Jacobs (Reviewer #1) (co-reviewed by Brianna Weiss); Marc Bramkamp (Reviewer #2); Carol Gross (Reviewer #3) (co-reviewed by Horia Todor).

Essential revisions:

1. Although the authors’ experimental design is fundamentally sound, there is the possibility of “jackpotting”, which could affect their results, particularly in the uninfected control experiment. If the authors’ Tn-seq library is ~200,000 strains, and they don’t plate at 10-100x fold access of colonies, then any given strain (regardless of its phenotype) may or may not be represented in the output of the experiment, causing false phenotypes to be ascribed to genes based on chance. This is particularly problematic for the uninfected control, where the authors choose to dilute the culture 1000-fold to mimic the number of colonies that survive infection. They may be better served by plating the whole culture on the plates, to ensure adequate representation of the library. Part of the reason for this concern is that an overwhelming majority of statistically significant hits (something like 80-90%) appear to confer susceptibility rather than resistance (source data Figure 2) – something the authors’ experimental design should not be able to measure. The lack of accurate representation of distributions of strains in the starting culture also calls into question the quantitative differences presented in the results.

a. L138. Where the authors describe their initial experimental design it would be helpful to add more details. What is the size of the Tn library? What is the coverage in their experiment? Approximately how many colonies are recovered on the plates after phage infection and in the uninfected control?

b. It is important to know how the number of colonies on the plates compares to the number of reads in the experiment. In the analysis of most HT screens, one implicitly assumes that each read corresponds to 1 cell, hence each read can be treated as statistically independent. This assumption is critical to the statistical methods used to analyze this data. By scraping a plate of colonies (which may be required for efficient phage infection), the authors potentially violate this assumption (since the number of cells → the number of colonies, are the actual statistically independent entities in the experiment). Does this assumption hold (or approximately hold) for the screen? If not, a different statistical method should be used to determine p-values.

2. The authors’ Tn-seq methodology is different from previously published HT-phage screens (e.g. Mutalik et al., 2020 and Rousset et al., 2018). Whilst it is clear that plating the infected cells has advantages, this rationale will not be clear for most people’performing such experiments. Please explain the rationale for the experimental protocol:

a. Why did the authors plate the cultures after initial phage absorption instead of allowing them to remain in liquid?

b. How reproducible are the authors’ Tn-seq results? The SRA ascension shows multiple replicates but this is not described in the manuscript nor reflected in the supplementary data. Given the potential for bottleneck and jackpotting effects in this assay, some measure of reproducibility is important for interpreting the results (see point 1).

c. L587 “Significant hits with fewer than 10 insertions on each strand were manually removed.” Why did the authors choose this criterion? Almost all of the genes they removed have very asymmetric distributions (e.g. in the Cog experiment, cgp3051 has 47853 fwd reads and 6 rev reads). Asymmetric distribution of insertions suggests that overexpression of downstream genes has an important (positive or negative) effect. This is a worthwhile pursuit, and many automated analysis pipelines can disambiguate these effects, including those developed in the Walker Lab (e.g. doi: 10.1038/s41589-018-0041-4). These genes shouldn’t be thrown away when they are arguably some of the most informative hits!

3. There is a somewhat extensive phylogeny of M. smegmatis phages (phagesdb.org). Are the phages that the authors work on related to any of these phages? If so, what cluster do they map to? What is the host range of other phages in that cluster? If not, may be worthwhile to mention that these are quite distinct from other studied phages.

4. Given that cgp_0475 was a strong hit in the Tn-seq, why was it not identified in the previous chemical genomics experiments from the lab (https://doi.org/10.7554/*eLife*.54761)?

5. Is there any relationship between the growth rate of the mutants and their phage susceptibility? This can be analyzed using the authors’ previous studies of this library.

6. The information on the localization of AhfA is sparse. In the discussion the authors speculate about a cytosolic localization, however, this is not proven.

7. In general, there were no p values relating to the statistical significance of any of the data presented, please address this (also highlighted in minor points below)

8. Figure 8 was not entirely supported by the data, especially Figure 8A which either could be improved with better images that support the claims.

*Reviewer #1 (Recommendations for the authors):*

– In Figure 1: the figure legend for section D, the phage adsorption assay, is mislabeled.

– Based on how Figures 1B and 1C are written, it would make more sense to switch these two figures in the final print.

– In the text explanation of Figure 4D, I would put (ECOI) after “Efficiency by which Cog formed centers of infection..” as it would make quick understanding of the Y axis of figure 4D easier.

– For figures 6C and D, I would quantify the percentage differences in phage adsorption between wild-type and knockout Cglu cells.

– Figure 8 could be strengthened with a better figure design, or images for 8A, as it did not entirely support the claims the authors made.

– In Figure 8: the figure legend for Figure 8C, TLC analysis of lipids.. is mislabeled and should say C, not D.

– What is the reason for using Cog and CL31 specifically beyond the fact that they infect Cglu?

– Line 79 should read “reversible”.

– Line 123 should read “possibly due to its role”.

– Line 124 should read “Sanger sequencing”.

– Line 125 should read “gene in the altered plaque”.

– Line 362 should read “reveal phage receptors”.

– Line 380 should read “ infection suggests that”.

– The SNPs in clg55 of the CP CL31s, are those the main differences you see in the CP CL31s vs wt? Anything else it could be?

– The CP CL31 with the clg55 SNPs is separate from the lab-adapted CL31 which shows similar plaque heterogeneity if I’m reading this right and the cause of the heterogeneity in the lab-adapted strain was not looked into. Why not look for those or similar SNPs?

– Do you know what about Cog and CL31 is different that would necessitate these different pathways? Obviously, they seem to bind different receptors, anything else? This may be out of the scope of your work, but I think knowing the phage differences that relate to different ways of exploiting the bacteria could be helpful in creating ways through the envelope.

– I mentioned these in public feedback but I’d like to see you address any changes in the lab strain (if you think there are any) that may have impacted results and also back up your significance of creating antibiotics with this info a little better.

*Reviewer #2 (Recommendations for the authors):*

Congratulations to both authors on this nice story!

The only extra experiment that I would like ’o suggest is the (sub-)cellular localization of AhfA. The authors speculate in the discussion about cytoplasmatic localization. It would be nice to really determine that by cell fractionation or imaging methods.

*Reviewer #3 (Recommendations for the authors):*

This is a well-presented manuscript with important conclusions, both about phage requirements and mycolic acid synthesis, with a significant follow-up of important hits.

---

## [Author Response]

Essential revisions:1. Although the authors’ experimental design is fundamentally sound, there is the possibility of “jackpotting”, which could affect their results, particularly in the uninfected control experiment. If the authors’ Tn-seq library is ~200,000 strains, and they don’t plate at 10-100x fold access of colonies, then any given strain (regardless of its phenotype) may or may not be represented in the output of the experiment, causing false phenotypes to be ascribed to genes based on chance. This is particularly problematic for the uninfected control, where the authors choose to dilute the culture 1000-fold to mimic the number of colonies that survive infection. They may be better served by plating the whole culture on the plates, to ensure adequate representation of the library. Part of the easonn for this concern is that an overwhelming majority of statistically significant hits (something like 80-90%) appear to confer susceptibility rather than resistance (source data Figure 2) – something the authors’ experimental design should not be able to measure. The lack of accurate representation of distributions of strains in the starting culture also calls into question the quantitative differences presented in the results.

We thank the reviewer for their thorough analysis of our experimental design. The Tn-Seq experiments were repeated with the uninfected controls plated at a density that maintains the representation of the original library. The overall results are largely unchanged because we maintain our focus on hits that become greatly enriched following phage infection not those that become depleted. The vast majority of these hits were validated for their involvement by constructing mutant strains, indicating the robustness of the current and previous analyses. With respect to the depletion of insertion mutants, we mentioned in the original submission that they are unlikely to be biologically meaningful.

a. L138. Where the authors describe their initial experimental design it would be helpful to add more details. What is the size of the Tn library? What is the coverage in their experiment? Approximately how many colonies are recovered on the plates after phage infection and in the uninfected control?

This information has been added (Figure 2 table supplement 1).

b. It is important to know how the number of colonies on the plates compares to the number of reads in the experiment. In the analysis of most HT screens, one implicitly assumes that each read corresponds to 1 cell, hence each read can be treated as statistically independent. This assumption is critical to the statistical methods used to analyze this data. By scraping a plate of colonies (which may be required for efficient phage infection), the authors potentially violate this assumption (since the number of cells → the number of colonies, are the actual statistically independent entities in the experiment). Does this assumption hold (or approximately hold) for the screen? If not, a different statistical method should be used to determine p-values.

We respectfully disagree with the reviewer on this point. In our view, a slurry of colonies from a plate is no different than a culture. Both contain a mixture of cells containing an array of different transposon mutants each represented multiple times in the population due to replication of the original mutant. We do not think there is any meaningful difference when it comes to the analysis whether this replication occurs in liquid or on a plate. In both cases, a read corresponds to a single cell/molecule of purified genomic DNA from the population.

2. The authors’ Tn-seq methodology is different from previously published HT-phage screens (e.g. Mutalik et al., 2020 and Rousset et al., 2018). Whilst it is clear that plating the infected cells has advantages, this rationale will not be clear for most people performing such experiments. Please explain the rationale for the experimental protocol:a. Why did the authors plate the cultures after initial phage absorption instead of allowing them to remain in liquid?

Although the authors in the Mutalik et al. paper did do competition experiments in liquid over several infection cycles, they also made use of a solid platebased assay in which they adsorbed their phages to the library cells for 15 minutes before plating. These plates were incubated overnight and resistant colonies were scraped, pelleted, and DNA prepped in a similar manner to the approach we took.

We prefer plating over liquid growth because colony formation is an easy way to ensure that the mutant population has undergone numerous rounds of doubling under a given condition before the analysis is performed.

b. How reproducible are the authors’ Tn-seq results? The SRA ascension shows multiple replicates but this is not described in the manuscript nor reflected in the supplementary data. Given the potential for bottleneck and jackpotting effects in this assay, some measure of reproducibility is important for interpreting the results (see point 1).

We performed completely new Tn-seq experiments for each phage in duplicate. The hit lists remained largely unchanged from our initial analysis and those that were previously selected for further analysis in the prior submission were also enriched for insertions in both new data sets. Thus, the results are highly reproducible.

c. L587 “Significant hits with fewer than 10 insertions on each strand were manually removed.” Why did the authors choose this criterion? Almost all of the genes they removed have very asymmetric distributions (e.g. in the Cog experiment, cgp3051 has 47853 fwd reads and 6 rev reads). Asymmetric distribution of insertions suggests that overexpression of downstream genes has an important (positive or negative) effect. This is a worthwhile pursuit, and many automated analysis pipelines can disambiguate these effects, including those developed in the Walker Lab (e.g. doi: 10.1038/s41589-018-0041-4). These genes shouldn’t be thrown away when they are arguably some of the most informative hits!

We have updated the criteria we used for selecting the most impactful insertion enrichments. Our concern in this report was to investigate mutants that affect phage infection when inactivated. We will pursue genes that affect phage infection when overexpressed (as indicated by asymmetric insertion orientation distributions) in a follow-on study. We think such a study would best be carried out with a different transposon containing a strong outward facing promoter.

3. There is a somewhat extensive phylogeny of M. smegmatis phages (phagesdb.org). Are the phages that the authors work on related to any of these phages? If so, what cluster do they map to? What is the host range of other phages in that cluster? If not, may be worthwhile to mention that these are quite distinct from other studied phages.

We agree that the phylogenetic history of corynephages is quite interesting. Very few phages that infect *Cglu* have been isolated and sequenced, let alone studied. Neither Cog nor CL31 share significant nucleotide identity with other sequenced phages and thus do not have assigned clusters at the moment.

4. Given that cgp_0475 was a strong hit in the Tn-seq, why was it not identified in the previous chemical genomics experiments from the lab (https://doi.org/10.7554/eLife.54761)?

We appreciate the reviewer’s interest in previous work from the lab. In the prior phenotypic analysis, *cgp_0475* was identified as having severe fitness defects across many conditions. However, it was not possible to correlate its phenotype with other genes involved in mycolic acid synthesis like *pks* and *fadD2* because they were found to be so sick in the phenotypic outgrowth that they were classified as essential.

5. Is there any relationship between the growth rate of the mutants and their phage susceptibility? This can be analyzed using the authors’ previous studies of this library.

While some of the phage resistant mutants are associated with poor fitness (namely those involved in mycolic acid synthesis), not all were associated with decreased growth. For example, there were minimal fitness defects associated with deletions of either *porAH* or the genes involved GalN decoration. However, loss of these genes greatly inhibited the ability of Cog to infect.

6. The information on the localization of AhfA is sparse. In the discussion the authors speculate about a cytosolic localization, however, this is not proven.

We have tried several ways of determining the precise localization of AhfA. Unfortunately, despite trying numerous tags on both the N- and C- termini, we have not been able to generate a functional tagged fusion. Therefore, further information on the localization of AhfA will have to await a follow-up study when we can find an appropriate fusion or generate antibodies against purified protein. Given that AhfA lacks a signal sequence and appears to be needed for TMM biogenesis, which is made on the cytoplasmic side of the membrane, we think it easonnable to predict a cytoplasmic localization for AhfA for now.

7. In general, there were no p values relating to the statistical significance of any of the data presented, please address this (also highlighted in minor points below)

We added the p-values as requested.

8. Figure 8 was not entirely supported by the data, especially Figure 8A which either could be improved with better images that support the claims.

We do not understand why the reviewer believes that Figure 8A does not support our conclusions. The mutant cells do not label with the 6-TMR-Tre dye whereas the WT control does. The dye labels mycolic acid such that our conclusion that AhfA is involved in mycolic acid synthesis is valid. In any case, we have included an additional supplementary source data file of the uncropped image of the 6-TMR-Tre treated cells to show a larger number of mutant cells that fail to stain, further supporting our conclusion.

Reviewer #1 (Recommendations for the authors):– In Figure 1: the figure legend for section D, the phage adsorption assay, is mislabeled.

The legend has been corrected

– Based on how Figures 1B and 1C are written, it would make more sense to switch these two figures in the final print.

We prefer to keep the order as is because it allows for the introduction of both the Cog and CL31 phages before getting into the details of the CL31 clear plaque variant.

– In the text explanation of Figure 4D, I would put (ECOI) after “Efficiency by which Cog formed centers of infection..." as it would make quick understanding of the Y axis of figure 4D easier.

Changed accordingly.

– For figures 6C and D, I would quantify the percentage differences in phage adsorption between wild-type and knockout Cglu cells.

Statistics have been added to determine if differences are significant.

– Figure 8 could be strengthened with a better figure design, or images for 8A, as it did not entirely support the claims the authors made.

We do not understand why the reviewer believes that Figure 8A does not support our conclusions. The mutant cells do not label with the 6-TMR-Tre dye whereas the WT control does. The dye labels mycolic acid such that our conclusion that AhfA is involved in mycolic acid synthesis is valid. In any case, a new supplementary figure was added showing a lager field of view to further support our conclusion.

– In Figure 8: the figure legend for Figure 8C, TLC analysis of lipids... is mislabeled and should say C, not D.

Corrected.

– What is the reason for using Cog and CL31 specifically beyond the fact that they infect Cglu?

CL31 and Cog were selected because they were the only *Cglu* phages available from the Félix d'Hérelle Reference Center for Bacterial Viruses. Unlike for phages of *M. smegmatis*, there are very few phages that infect *Cglu* that have been reported or available at stock centers.

– Line 79 should read "reversible".

Changed.

– Line 123 should read "possibly due to its role".

Changed.

– Line 124 should read "Sanger sequencing".

Changed.

– Line 125 should read "gene in the altered plaque".

Changed.

– Line 362 should read "reveal phage receptors".

Changed.

– Line 380 should read " infection suggests that".

Changed.

– The SNPs in clg55 of the CP CL31s, are those the main differences you see in the CP CL31s vs wt? Anything else it could be?

The SNPs in clg55 are the main differences. Several different rounds of WGS have not implicated other SNPs causing the difference.

– The CP CL31 with the clg55 SNPs is separate from the lab-adapted CL31 which shows similar plaque heterogeneity if I'm reading this right and the cause of the heterogeneity in the lab-adapted strain was not looked into. Why not look for those or similar SNPs?

The SNP was the cause of the heterogeneity. Sequencing of spontaneous new clear plaque CL31 isolates only implicates SNPs in clg55 as the difference between the phages. Also, the clear plaque variants only give rise to a homogeneous population of plaques.

– Do you know what about Cog and CL31 is different that would necessitate these different pathways? Obviously, they seem to bind different receptors, anything else? This may be out of the scope of your work, but I think knowing the phage differences that relate to different ways of exploiting the bacteria could be helpful in creating ways through the envelope.

We agree with the reviewer that understanding the differences between phages to predict receptor choice and injection mechanisms would be of interest. However, such an investigation is better done when we have a much larger collection of *Cglu* phages and can make more accurate predictions.

– I mentioned these in public feedback but I'd like to see you address any changes in the lab strain (if you think there are any) that may have impacted results and also back up your significance of creating antibiotics with this info a little better.

The lab strain has a large number of changes relative to the environmental strains. None of the strains are in the same lineage (i.e. the lab strain was not derived from the environmental strains used as the original phage hosts). Thus, the large number of differences in their genomes is not surprising. Obviously, results with the phage infections will be strain dependent as is typical given the specificity phages have for their hosts. However, the general messages of our study with the MB001 lab strain regarding envelope structure and infection requirements are likely to be relevant for other members of Corynebacteriales and their phages.

Reviewer #2 (Recommendations for the authors):Congratulations to both authors on this nice story!The only extra experiment that I would like to suggest is the (sub-)cellular localization of AhfA. The authors speculate in the discussion about cytoplasmatic localization. It would be nice to really determine that by cell fractionation or imaging methods.

We have tried several ways of determining the precise localization of AhfA. Unfortunately, despite trying numerous tags on both the N- and C- termini, we have not been able to generate a functional tagged fusion. Therefore, further information on the localization of AhfA will have to await a follow-up study when we can find an appropriate fusion or generate antibodies against purified protein. Given that AhfA lacks a signal sequence and appears to be needed for TMM biogenesis, which is made on the cytoplasmic side of the membrane, we think it is reasonable to predict a cytoplasmic localization for AhfA for now.